# Stn1 supports Mec1 function in protecting stalled replication forks from degradation

Erika Casari🔴⊕, Flavio Corallo🔴⊕, Luca Edoardo Milani🔴, Renata Tisi,
Maria Pia Longhese🔴*

Dipartimento di Biotecnologie e Bioscienze, Università degli Studi di Milano-Bicocca, Milano, Italy

⊕ These authors contributed equally to this work
* mariapia.longhese@unimib.it

## Abstract

Replication stress threatens genome integrity by exposing replication forks to nucleo-lytic degradation. In both yeast and humans, the checkpoint kinases Mec1 and Rad53 limit deleterious single-stranded DNA (ssDNA), yet the protective mechanisms remain incompletely defined. Here, we identify a role for the CST subunit Stn1 in cooperating with Mec1 to restrain ssDNA formation under nucleotide depletion. A gain-of-function allele (*stn1-L60F*) suppresses the sensitivity to replication stress of Mec1-deficient cells and reduces ssDNA at stalled replication forks, whereas a loss-of-function trun-cation (*stn1-ΔC*) exacerbates both phenotypes. Mechanistically, Stn1 opposes the resection activities of Mre11, Exo1, and Sgs1 by promoting Polα-primase-dependent fill-in and by limiting their association with stalled replication forks, with the latter mech-anism predominating in the suppression exerted by Stn1$^{L60F}$. Thus, Stn1 works with the checkpoint to curb nuclease activity at sites of replication stress.

## Author summary

Each time a cell divides, it must accurately duplicate its DNA. This complex process can be disrupted by stress or DNA damage, compromising the replica-tion fork, the structure where DNA replication takes place. Unprotected forks can collapse, leading to genome instability, a hallmark of cancer and other diseases. In this study, we discovered that Stn1, a protein of the CST complex, plays a critical role in protecting stalled replication forks from degradation, especially when the main DNA damage checkpoint protein Mec1 is not fully functional. Stn1 prevents the accumulation of single-stranded DNA (ssDNA) by limiting nucleas-es, the enzymes that degrade DNA, from excessively resecting replication forks. Remarkably, we identified a specific mutation in Stn1 (L60F) that enhances this protective function. Our findings uncover a role for Stn1 in safeguarding genome stability by acting as a backup to checkpoint pathways to control DNA processing at stressed forks.

**Data availability statement:** All relevant data are within the manuscript and its Supporting Information files.

**Funding:** This work was supported by Fondazione AIRC under IG 2022 - ID. 27001 project to M.P.L. and the European Union - Next Generation EU, Mission 4 Component 2 - CUP H53D23004770006 to M.P.L. The funders had no role in the study design, data collection and analysis, decision to publish or preparation of the manuscript.

**Competing interests:** The authors have declared that no competing interests exist.

## Introduction

During DNA replication, fork progression can be hindered by various obstacles, including DNA lesions, DNA secondary structures, nucleotide shortage, ribonucleotide incorporation, DNA:RNA hybrids (R-loops), and accumulation of topological stress [1]. These perturbations can disrupt the coordination between leading- and lagging-strand synthesis, or uncouple DNA polymerases from the helicase machinery, leading to the accumulation of single-stranded DNA (ssDNA) [1,2]. This ssDNA is rapidly coated by the replication protein A (RPA) complex, which promotes recruitment of the checkpoint kinase Mec1 [3,4]. Mec1, the yeast homolog of mammalian ATR, then activates the effector kinase Rad53 via the mediator Mrc1 [5,6]. Mec1 and Rad53 stimulate deoxyribonucleotide triphosphate (dNTP) production [7], inhibit activation of late-firing origins [8–10], and promote restart of stalled replication forks as well as replication through damaged templates [11–13].

Replication stress can be experimentally induced by the ribonucleotide reductase (RNR) inhibitor hydroxyurea (HU), which causes replication fork stalling by depleting intracellular dNTP pools. In budding yeast, fork stalling in the absence of a functional checkpoint leads to replication fork collapse, accumulation of abnormal replication intermediates, elevated levels of ssDNA, and cell death [2,11–16]. Similar phenomena have been observed in ATR-defective *Schizosaccharomyces pombe* and mammalian cells, where replication stress results in ssDNA accumulation in a manner dependent on MRE11 and EXO1 [17,18]. Restoring checkpoint function after fork arrest does not rescue viability [13], indicating that checkpoint signaling must act at the time of fork arrest to prevent an irreversible collapse of replication forks.

How the checkpoint prevents irreversible fork collapse and promotes cell survival remains poorly understood. Rad53 inhibits origin firing by phosphorylating the firing factors Dbf4 and Sld3 [10]. However, a non-phosphorylatable *sld3 dbf4* double mutant does not display increased sensitivity to replication stress [10], suggesting that inhibition of origin firing is not essential for viability. While the checkpoint does not appear to be required for maintaining replication factor complexes at stalled forks [19,20], it has been suggested to regulate replisome progression following inhibition of DNA synthesis to prevent helicase-polymerase uncoupling [21–25]. Consistent with such a role, Rad53 phosphorylates Mrc1 and Mcm10, preventing their stimulation of DNA unwinding by the CMG helicase [26]. Moreover, the helicases Rrm3 and Pif1 are phosphorylated in a Rad53-dependent manner following replication stress [27]. A recent in vitro reconstitution of replication stalling and restart in budding yeast under nucleotide depletion showed that leading-strand DNA synthesis halts, while the CMG helicase continues to unwind and Okazaki fragments continue to initiate on the lagging strand [28]. The resulting incomplete Okazaki fragments sequester essential replication factors such as PCNA, RFC, and DNA polymerases δ and ε, thus preventing the resumption of processive DNA synthesis and leaving nascent DNA vulnerable to nuclease attack [28,29]. The checkpoint counteracts this process by inhibiting CMG unwinding, preventing excessive Okazaki fragment accumulation and protecting stalled forks from nuclease-mediated degradation [28,29].

The action of nucleases can be restricted by several factors. In mammalian cells, BRCA1 and BRCA2 protect nascent DNA from degradation by the MRE11 nuclease [30], whereas BOD1L protects DNA from the DNA2-WRN nuclease-helicase complex [31]. Furthermore, human Rif1 was shown to protect the nascent DNA from DNA2-WRN-mediated degradation in a manner that depends on its interaction with the PP1 phosphatase [32,33].

Among the proteins that limit ssDNA accumulation is the highly conserved CST complex. In *Saccharomyces cerevisiae*, CST is composed of Cdc13, Stn1, and Ten1, whereas in humans it consists of CTC1, STN1, and TEN1. Structurally related to the RPA complex, CST binds both ssDNA and ssDNA-double-strand DNA (dsDNA) junctions [34–38]. In both budding yeast and human cells, CST plays essential roles at telomeres, terminating G-strand elongation by telomerase and promoting C-strand synthesis by stimulating the Polα-primase complex [39–46]. In both *S. cerevisiae* and *S. pombe*, CST also protects telomeres from Exo1-mediated degradation and contributes to chromosome stability [47–55].

Beyond telomeres, in *S. pombe* the CST complex promotes replication fork progression at repetitive sequences such as subtelomeres and ribosomal DNA (rDNA) [56,57], whereas in human cells CST localizes to stalled replication forks and promotes genome-wide replication restart after HU treatment [58–60]. More recently, human CST has been shown, on the one hand, to antagonize DNA end resection at DNA double-strand breaks (DSBs) by recruiting the Polα-primase complex and promoting fill-in synthesis under the regulation of 53BP1, RIF1, and shieldin [61–64]. In *S. cerevisiae*, CST likewise limits resection at a Cas9-induced DSB through its interaction with the Polα-primase complex, preventing a mutational signature associated with non-homologous end joining (NHEJ) [65]. On the other hand, other studies have reported that CST directly blocks degradation of nascent DNA by the MRE11 nuclease [66], and in vitro CST directly suppresses the resection activity of EXO1 and the BLM-DNA2 helicase-nuclease complex [67]. How these protective functions of CST are mechanistically coordinated with the checkpoint response remains to be determined.

Here, we show that the *S. cerevisiae* protein Stn1 plays a critical role in restraining nucleolytic degradation of stalled replication forks, especially when Mec1 is partially compromised. The gain-of-function *stn1-L60F* allele suppresses the HU sensitivity of cells carrying the hypomorphic *mec1-100* allele, whereas the loss-of-function *stn1-ΔC* allele exacerbates it. The severe HU sensitivity of *mec1-100 stn1-ΔC* cells correlates with increased ssDNA at stalled forks, while ssDNA is reduced in *stn1-L60F mec1-100* cells. ssDNA accumulation in *stn1-ΔC mec1-100* cells depends on Mre11, Exo1, and Sgs1, indicating that Stn1 antagonizes these resection activities. Stn1 exerts this function by promoting Polα-primase-dependent fill-in synthesis and by limiting the association of Mre11, Exo1 and Sgs1 with stalled replication forks as well as with DNA DSBs. Together, these findings indicate that Stn1 supports Mec1 function in protecting sites of replication stress from nuclease-mediated degradation, establishing a functional link between CST activity and checkpoint control.

## Results

### The *stn1-L60F* allele suppresses the HU sensitivity of *mec1-100* cells, whereas *stn1-ΔC* exacerbates it

To investigate the pathways that support Mec1 during DNA replication under stress conditions, we performed a genetic screen to identify extragenic mutations that suppress the HU sensitivity of Mec1-deficient cells. Because *mec1*-null cells (viable only in the absence of *SML1*) are extremely sensitive to HU and undergo extensive replication fork degradation even at low HU doses [12,15], we employed the hypomorphic *mec1-100* allele [68]. This allele confers milder HU sensitivity and is defective in the intra-S checkpoint, whereas the G2/M checkpoint remains largely intact [13,68,69]. Moreover, in HU-treated *mec1-100* cells, lagging-strand DNA synthesis proceeds much farther than leading-strand synthesis, exposing long stretches of single-stranded leading-strand templates [70].

From this screen, we isolated HU-resistant *mec1-100* clones that fell into four distinct allelism groups. Genetic analysis and whole-genome sequencing revealed suppressor mutations in four genes: *SRB8*, encoding a subunit of the transcriptional Mediator complex; *RPS5*, encoding a ribosomal protein; *PPH3*, encoding the catalytic subunit of the PP4 phosphatase; and *STN1*, encoding a subunit of the CST complex. As loss of *PPH3* has been shown to suppress *mec1-100* HU

sensitivity by restoring phosphorylation of Rad53 and other checkpoint targets [71], and mutations in *SRB8* or *RPS5* may cause pleiotropic effects that complicate mechanistic dissection, we focused on *STN1*.

The *stn1* suppressor allele contains a leucine-to-phenylalanine substitution at position 60 (*stn1-L60F*), located within the oligonucleotide/oligosaccharide-binding (OB)-fold domain of Stn1 [72]. While *STN1* is essential for cell viability, deletion of its C-terminal domain (residues 282–495) partially impairs Stn1 function by disrupting its interaction with both Cdc13 and Pol12 (a subunit of the Polα-primase complex), leading to telomere over-elongation with minimal growth defect [42,43,73]. To assess whether *stn1-L60F* suppresses *mec1-100* via a loss-of-function mechanism, we introduced the *stn1-ΔC* allele into *mec1-100* cells. Unlike *stn1-L60F mec1-100*, *stn1-ΔC mec1-100* cells displayed increased HU sensitivity (Fig 1A and 1B), indicating that suppression by *stn1-L60F* does not result from reduced Stn1 function.

While the increase in HU sensitivity of *mec1-100* cells conferred by *stn1-ΔC* was recessive (Fig 1C), suppression by the *stn1-L60F* allele was dominant to both wild-type *STN1* and *stn1-ΔC*, as *STN1/stn1-L60F mec1-100/mec1-100* and *stn1-L60F/stn1-ΔC mec1-100/mec1-100* diploid cells showed reduced HU sensitivity compared with *STN1/STN1 mec1-100/mec1-100* diploid cells (Fig 1C). Moreover, *stn1-L60F* does not compensate for complete loss of Mec1, as it fails to suppress the markedly more severe HU sensitivity of *mec1Δ* cells (kept viable by *SML1* deletion) compared with *mec1-100* (S1A Fig). The *stn1-L60F* mutation did not alter protein levels, as similar amounts of Stn1 were detected in protein extracts from wild-type and *stn1-L60F* cells (Fig 1D). Finally, *stn1-L60F* did not affect telomere length, while, as expected, the *stn1-ΔC* allele led to telomere over-elongation (S1B Fig) [42,43,73].

## The *stn1-L60F* allele decreases checkpoint activation in *mec1-100* cells, whereas *stn1-ΔC* increases it

Rad53 activation requires its phosphorylation, which is detected as reduced electrophoretic mobility. We previously showed that *mec1-100* cells exhibit delayed Rad53 phosphorylation in response to replication stress [68]. When cells were released from G1 arrest into HU-containing medium, Rad53 phosphorylation was detected immediately in wild-type cells, but was markedly delayed in *mec1-100* mutants (Fig 1E and 1F). This delay was suppressed in *mec1-100 stn1-ΔC* cells, whereas it was exacerbated in *mec1-100 stn1-L60F* cells (Fig 1E and 1F). A similar pattern was observed upon treatment with phleomycin, a DNA-damaging agent that induces checkpoint activation (Fig 1G and 1H). The *stn1-L60F* mutation alone caused a mild reduction in Rad53 phosphorylation compared with wild-type cells after both HU (Fig 1E and 1F) and phleomycin treatment (Fig 1G and 1H), consistent with impaired checkpoint activation. Because Mec1-dependent Rad53 phosphorylation is triggered by the generation of ssDNA, these findings suggest that *stn1-ΔC* enhances checkpoint signaling by increasing ssDNA formation at stalled replication forks, whereas *stn1-L60F* limits ssDNA accumulation, resulting in attenuated checkpoint activation. In any case, as *stn1-L60F* reduces Rad53 phosphorylation in *mec1-100* cells, the increased HU resistance of *mec1-100 stn1-L60F* cells cannot be ascribed to restored checkpoint activity.

## The *stn1-L60F* allele decreases ssDNA generation at stalled replication forks in *mec1-100* cells, whereas *stn1-ΔC* increases it

We previously showed that the HU sensitivity of *mec1-100* cells can be exacerbated by increasing the generation of ssDNA at stalled replication forks upon deletion of the resection inhibitor Rad9 [74]. To directly quantify ssDNA at stalled forks, we employed a quantitative PCR (qPCR)-based assay that exploits the resistance of ssDNA to cleavage by the restriction enzyme SspI, which cuts only dsDNA. Genomic DNA from both SspI-digested and mock-digested samples was amplified with primers flanking SspI sites, and products were normalized to a control amplicon located on chromosome XI.

Cells were synchronized in G1 with α-factor and released into HU-containing medium. We then measured ssDNA by qPCR at increasing distances from the early-firing origin ARS607. Previous studies have shown that in untreated wild-type cells, ssDNA spans approximately 220 nucleotides, likely reflecting the region engaged by the replisome [2]. HU increases the length of these gaps because helicase unwinding becomes uncoupled from DNA polymerases [75–77].

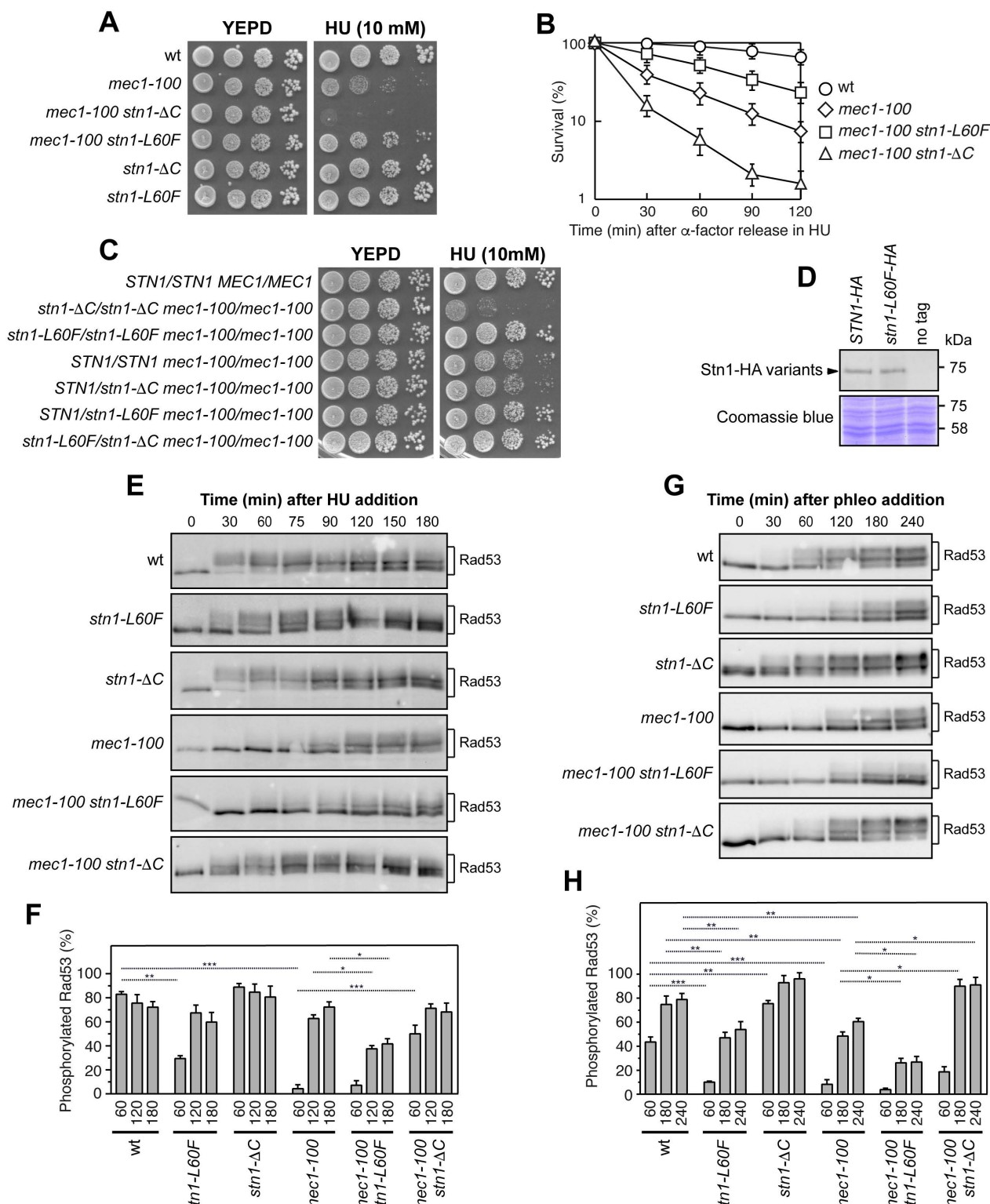

**Fig 1. Opposite effects of *stn1-L60F* and *stn1-ΔC* on the HU sensitivity and checkpoint activation of *mec1-100* cells. (A)** Exponentially growing cell cultures were serially diluted (1:10) and each dilution was spotted out onto YEPD plates with or without HU. **(B)** Cells were arrested in G1 with α-factor (time zero) and then released into YEPD containing 0.2M HU. Aliquots were removed from the HU-treated cultures at timed intervals to score

for colony-forming units on YEPD plates at 25°C. Plotted values are the mean values±s.d. from three independent experiments. **(C)** Exponentially growing cell cultures were serially diluted (1:10) and each dilution was spotted out onto YEPD plates with or without HU. **(D)** Western blot analysis with an anti-HA antibody of protein extracts prepared from exponentially growing cells. The same amount of extracts was stained with Coomassie Blue as loading control. **(E)** Cells were arrested in G1 with α-factor (time zero) and then released into YEPD containing 0.2M HU. Protein extracts prepared at different time points after α-factor release were analyzed by western blot using an anti-Rad53 antibody. **(F)** Quantitative analysis of Rad53 phosphorylation shown in panel (E) was performed by calculating the ratio of band intensities for slowly-migrating bands to the total amount of protein. Plotted values are the mean values±s.d. from three independent experiments. ***$p < 0.005$, **$p < 0.01$, *$p < 0.05$ (Student's $t$-test). **(G)** Phleomycin (10 μg/mL) was added to exponentially growing cells, and protein extracts were analyzed by western blot using an anti-Rad53 antibody. **(H)** Quantitative analysis of Rad53 phosphorylation shown in panel (G) was performed as in panel **(F)**.

As expected, HU-treated wild-type cells accumulated ssDNA above background levels (as defined by α-factor arrest) near ARS607, with a progressive decrease at greater distances (Fig 2). Both the amount and the extent of ssDNA were higher in *mec1-100* cells (Fig 2), consistent with the accumulation of single-stranded leading-strand template due to uncoupled leading- and lagging-strand synthesis [70]. This ssDNA accumulation remained confined to regions near the replication origin, as no significant change was detected at a distal control locus (Fig 2). A similar ssDNA increase was observed in HU-treated *stn1-ΔC* cells. Notably, the *stn1-L60F* allele reduced ssDNA in *mec1-100* cells to wild-type levels, whereas *mec1-100 stn1-ΔC* double mutants increased both the amount and the extent of ssDNA relative to *mec1-100* alone (Fig 2). These findings indicate that Stn1 limits ssDNA formation at stalled replication forks in *mec1-100* cells, and that *stn1-L60F* further strengthens this protective effect.

### Suppression of Mec1 deficiency by *stn1-L60F* depends on inhibition of Exo1, Sgs1 and Mre11

Mre11, Exo1, and Dna2 have been implicated in processing stalled replication forks and in preventing the accumulation of aberrant DNA intermediates in *rad53* mutants [14,16]. To test whether these nucleases are involved in the Stn1-dependent limitation of ssDNA, we deleted *EXO1* or introduced the nuclease-deficient *mre11-H125N* allele into *mec1-100* cells. Because Dna2 is essential for cell viability, we deleted the helicase-encoding gene *SGS1*, which is required to support Dna2 during DSB resection [78–81].

Deletion of *EXO1* or *SGS1*, as well as expression of *mre11-H125N*, slightly increased the HU sensitivity of *mec1-100* cells (Figs 3 and 4), suggesting that a limited level of ssDNA formation is beneficial for coping with replication stress. Moreover, *exo1Δ*, *sgs1Δ*, and *mre11-H125N* were epistatic to both *stn1-ΔC* and *stn1-L60F*: the HU sensitivity of *mec1-100 exo1Δ*, *mec1-100 sgs1Δ*, and *mec1-100 mre11-H125N* cells was unchanged upon introduction of *stn1-ΔC* (Fig 3), and *stn1-L60F* failed to suppress the HU sensitivity of *mec1-100 mre11-H125N*, *mec1-100 exo1Δ* and *mec1-100 sgs1Δ* cells (Fig 4). These findings indicate that the deleterious *stn1-ΔC* phenotype is mediated by Exo1, Sgs1, and Mre11, and that *stn1-L60F* fails to suppress in *exo1Δ*, *sgs1Δ*, or *mre11-H125N* background because loss of any single nuclease overrides its beneficial effect. Notably, the involvement of both Sgs1 and Exo1 in processing stalled replication forks differs from their apparently redundant roles in the resection of broken DNA ends, where deletion of either gene alone results in only minor defects [78,79].

We then quantified ssDNA at ARS607 by qPCR upon release from G1 into HU-containing medium. Deletion of *EXO1* or *SGS1*, or expression of *mre11-H125N*, strongly reduced ssDNA accumulation in *mec1-100* (S2 Fig) and *mec1-100 stn1-ΔC* cells (Fig 5), indicating that formation of ssDNA in *mec1-100* cells depends on Mre11, Exo1, and Sgs1. Because removing any one of these nucleases both increases HU sensitivity of *mec1-100* and abolishes *stn1-L60F*-mediated suppression, our data support the view that Stn1 restrains excessive nuclease engagement at stalled forks while a basal nuclease activity remains beneficial under HU treatment.

### The L60F mutation increases Stn1 interaction with DNA

The CST complex binds both ssDNA and ssDNA-dsDNA junctions [34–38]. Upon HU treatment, Stn1 was recruited to the early-firing origins ARS305 and ARS607, as assessed by ChIP-qPCR (Fig 6A). Although the L60F substitution did not

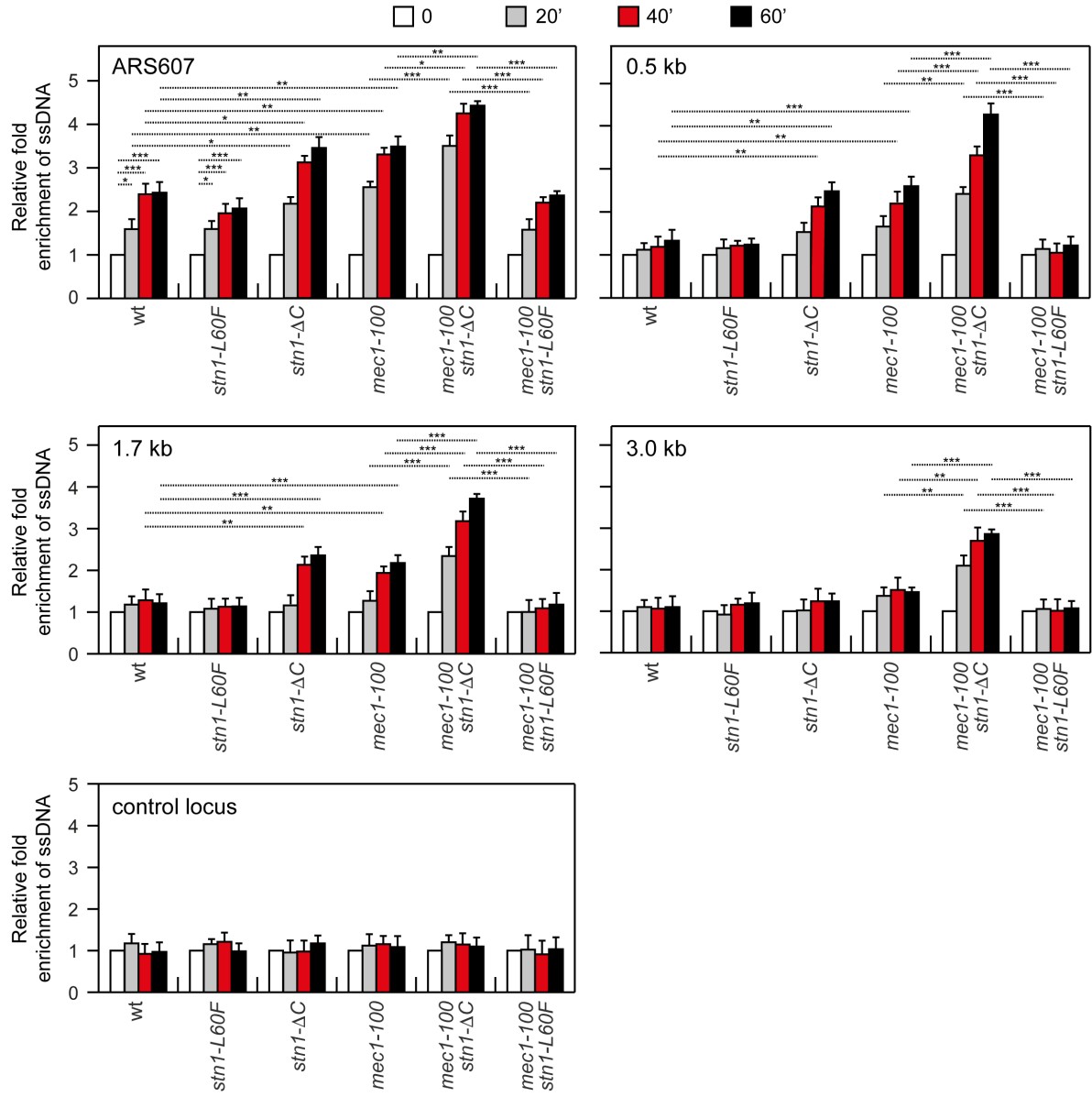

**Fig 2. Analysis of ssDNA at different distances from ARS607.** Exponentially growing YEPD cell cultures were arrested in G1 with α-factor (time zero) and then released into YEPD containing 0.2M HU. Genomic DNA prepared at different time points after α-factor release was either digested or mock-digested with SspI and used as a template in qPCR. The value of SspI-digested over non-digested DNAs was determined for each time point after normalization to an amplicon on chromosome XI that does not contain SspI sites. The data shown are expressed as fold-enrichments in ssDNA at different time points after α-factor release in HU relative to the α-factor (time zero) (set to 1.0). A locus containing SspI sites on chromosome XI is used as a control (control locus). Plotted values are the mean values ± s.d. from three independent experiments. ***$p<0.005$, **$p<0.01$, *$p<0.05$ (Student's $t$-test).

alter Stn1 protein abundance (Fig 1D), Stn1[L60F] showed higher occupancy at ARS305 and ARS607 than wild-type Stn1 (Fig 6A), indicating that L60F increases Stn1 association with DNA at stalled forks. Stn1 recruitment at HU-stalled forks did not require the Ku complex, which is itself recruited to arrested replication forks [82,83]. In fact, binding of both Stn1 and Stn1-L60F was unchanged in *ku70Δ* cells (Fig 6A).

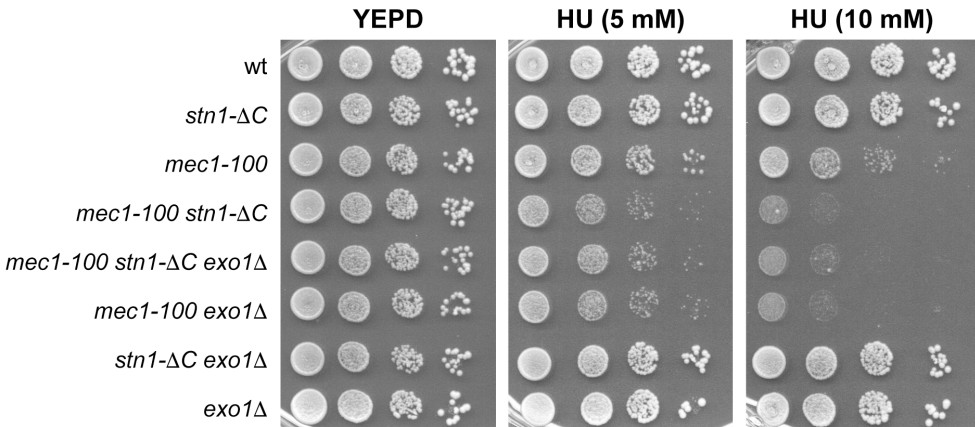

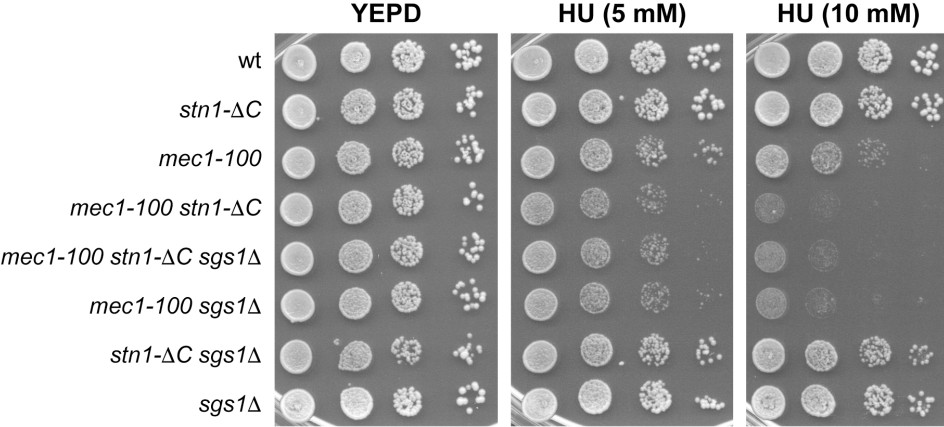

**Fig 3. *mre11-H125N*, *exo1Δ* and *sgs1Δ* are epistatic to *stn1-ΔC* with respect to the HU sensitivity of *mec1-100* cells.** Exponentially growing cell cultures were serially diluted (1:10) and each dilution was spotted out onto YEPD plates with or without HU.

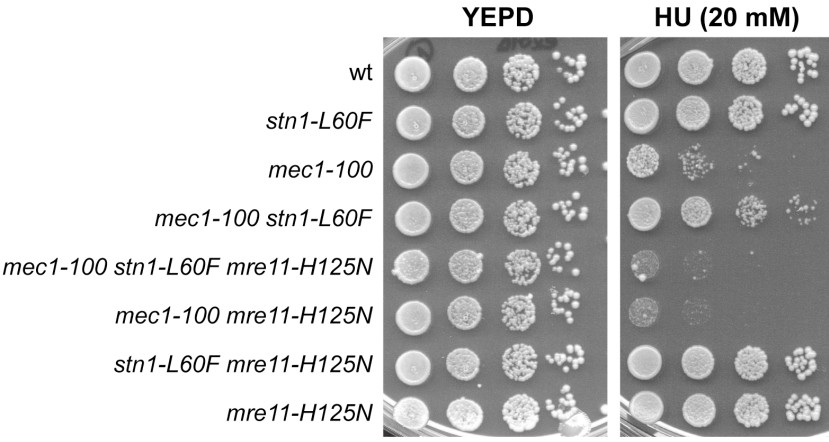

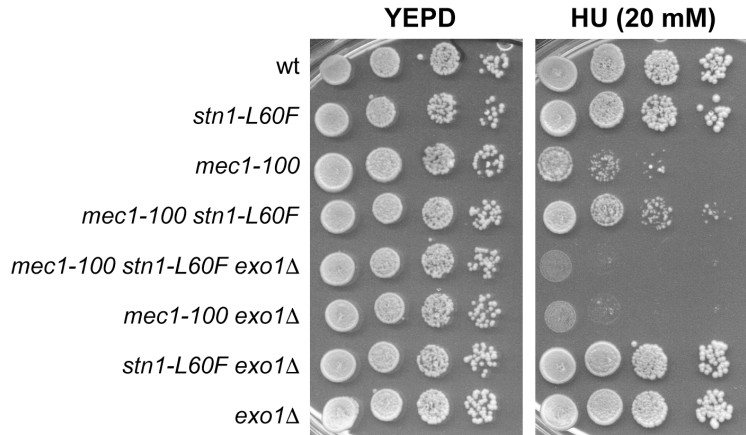

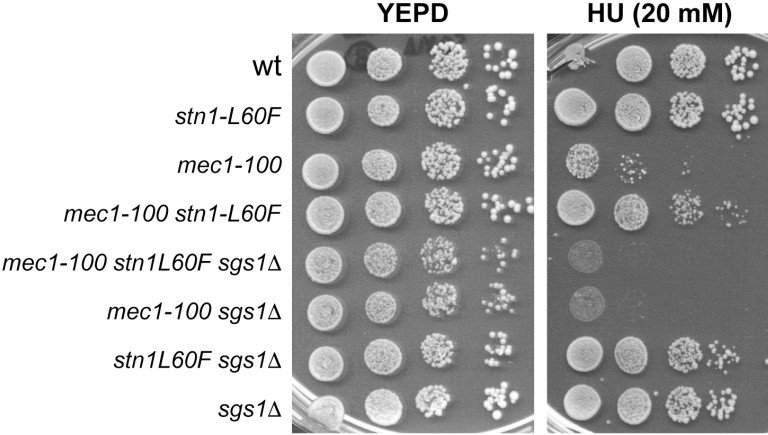

**Fig 4.** **mre11-H125N**, **exo1Δ** and **sgs1Δ** are epistatic to **stn1-L60F** with respect to the HU sensitivity of **mec1-100** cells. Exponentially growing cell cultures were serially diluted (1:10) and each dilution was spotted out onto YEPD plates with or without HU.

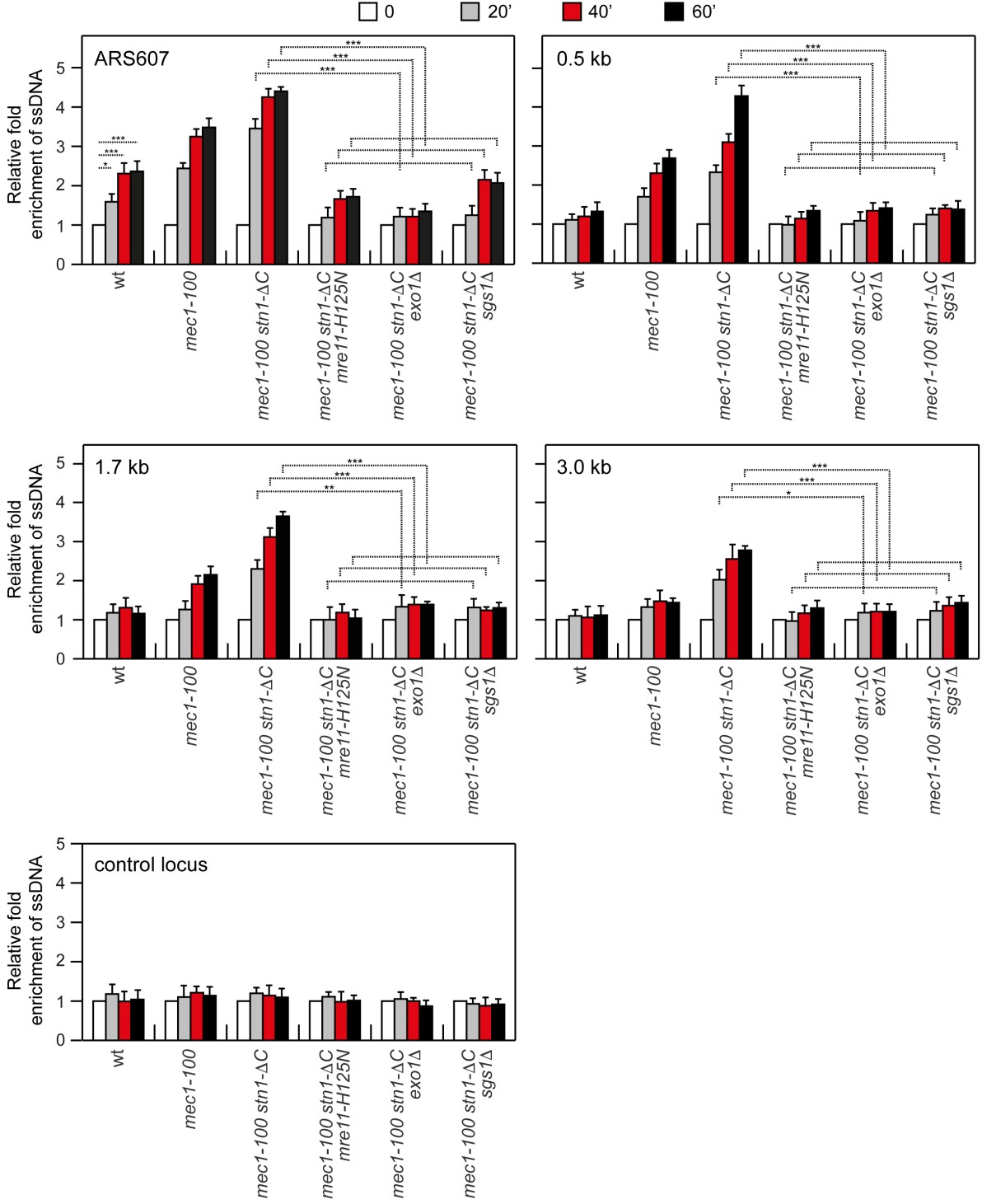

**Fig 5. ssDNA generation in HU-treated *mec1-100 stn1-ΔC* depends on Mre11 nuclease, Exo1, and Sgs1.** Exponentially growing YEPD cell cultures were arrested in G1 with α-factor (time zero) and then released into YEPD containing 0.2M HU. ssDNA at different distances from ARS607 was assessed as described in Fig 2. \*\*\*$p < 0.005$, \*\*$p < 0.01$, \*$p < 0.05$ (Student's *t*-test).

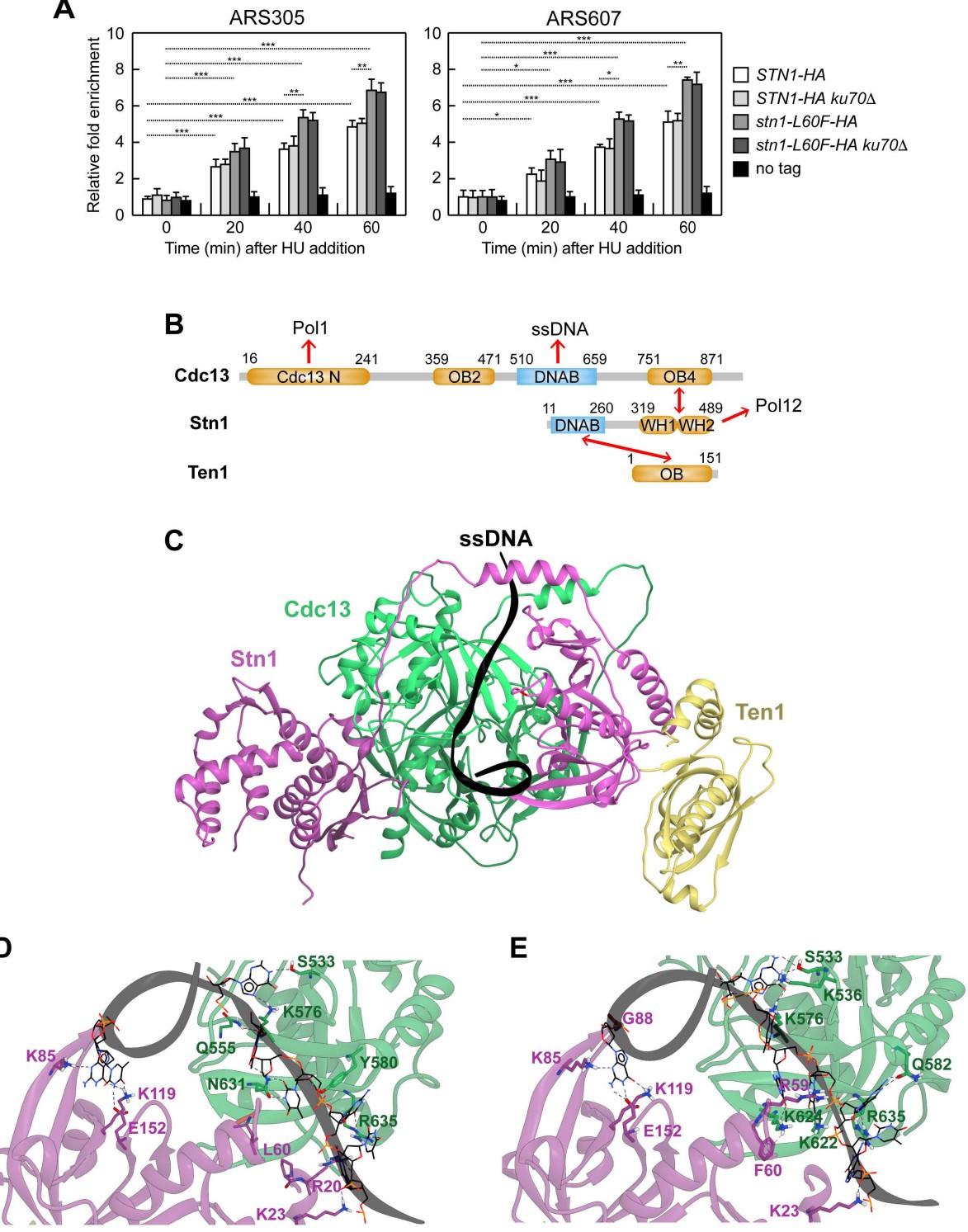

**Fig 6. The L60F mutation confers to the CST complex higher affinity for ssDNA. (A)** ChIP analysis of Stn1-HA and Stn1-L60F-HA. Cells were arrested in G1 with α-factor and then released into YEPD containing 0.2M HU at time zero. Relative fold-enrichment of HA-tagged Stn1 and Stn1-L60F at ARS305 and ARS607 replication origins was determined after ChIP with an anti-HA antibody and qPCR analysis. Plotted values are the mean values ± s.d. from three independent experiments. ***$p < 0.005$, **$p < 0.01$, *$p < 0.05$ (Student's *t*-test). **(B)** Domain architecture of CST components. Cdc13

contains four OB-fold domains: Cdc13-terminal (Cdc13 N), which binds Pol1, OB2, the DNA-binding domain or OB3 (DNAB), and OB4, which binds Stn1. Stn1 comprises an N-terminal OB-fold domain (DNAB), which binds DNA, and two C-terminal winged helix-turn-helix motifs (WH1 and WH2). The C-terminal region of Stn1 interacts with Cdc13 (via WH2) and with Pol12 (via WH1). Ten1 consists of a single OB-fold domain that binds the N-terminal OB-fold/DNA-binding domain of Stn1. Red arrows indicate protein-protein interactions. The domains are in orange except for the DNA-binding domains, in blue. **(C)** AlphaFold 3-predicted model of the yeast CST complex bound to a 20-nt ssDNA (black ribbon). The Cdc13 N domain is not depicted because its position is variable with respect to the rest of the complex (S3 Fig). **(D)** Detail of the ssDNA-binding interface in the HADDOCK 2.4 water refinement top-scoring model for wild-type CST complex. **(E)** Detail of the ssDNA-binding interface in the HADDOCK 2 water-refined top-scoring model for the mutant CS^L60F^T complex.

Both Cdc13 and Stn1 can contact DNA within the CST complex [34–38]. To provide a structural rationale for the effect of the L60F mutation, we used AlphaFold 3 predictor to model the CST complex bound to ssDNA (S3 Fig). In detail, a G/C-rich sequence was chosen according to the yeast telomeric G-tail consensus, a known substrate for CST binding [84]. AlphaFold 3 predicted several complexes, and the top-scoring model satisfied known relationships among CST protein domains [84]: ssDNA was contacted by the N-terminal domain of Stn1 and by the Cdc13 DNA-binding domain (OB3); the C-terminal OB-fold domain of Cdc13 contacted the C terminus of Stn1; the N-terminal OB domain of Stn1 also interacted with Ten1 (Fig 6B and 6C).

In the wild-type CST complex, both Stn1 and Cdc13 form multiple hydrogen bonds with ssDNA. The mutant complex was generated by introducing the L60F substitution into the wild-type model, and both models were refined with the HADDOCK 2.4 water-refinement protocol to optimize the contact interface. The protocol output comprises four models for each system and binding-energy estimates for the overall system (S1 Table). Compared with wild-type CST, the CS^L60F^T complex showed a lower (more favorable) energy content on average (S1 Table). To understand the source of this improvement, we analyzed the contact interface of the top-scoring models of each system. For instance, in the top-scoring wild-type model, Stn1 contacts ssDNA through K23, R20, K85, K119, and E152 residues, whereas Cdc13 contacts ssDNA through S533, Q555, K576, Y580, N631, and R635 residues (Fig 6D). Across the four models, the wild-type CST complex is predicted to form 16 protein-DNA hydrogen bonds on average, whereas the mutant CS^L60F^T complex formed 20 on average. The increase in the number of contacts reflects both higher persistence of the bonds described above over the different models and the appearance of new interactions. For example, Cdc13 contributed novel contacts such as K536, K622, and K624. A plausible explanation of this rearrangement is that replacing leucine with phenylalanine at position 60 in Stn1 enables π-cation interactions with the opposing Cdc13 interface, particularly K624, subtly reorienting the interface and increasing overall CST-DNA engagement (Fig 6E).

### Stn1 limits ssDNA at stalled forks by promoting Polα-primase-dependent fill-in and by restricting Mre11, Exo1, and Sgs1 association

The CST complex limits resection at DNA DSBs by recruiting the Polα-primase complex and promoting fill-in synthesis in both yeast and human cells [61–65]. CST has also been reported to protect stalled replication forks by blocking MRE11-dependent degradation of nascent DNA in human cells [66], to suppress resection by EXO1 and by the BLM-DNA2 helicase-nuclease complex [67], to antagonize Exo1-mediated resection at telomeres [47–55], and to support DNA replication at subtelomeres and rDNA regions [56,57]. On this basis, Stn1 could limit ssDNA at stalled replication forks by promoting Polα-primase-dependent fill-in of nascent gaps and/or by restricting the fork-proximal recruitment of Mre11, Exo1, and Dna2-Sgs1.

To assess the contribution of Polα-primase-mediated fill-in to limiting ssDNA accumulation in *mec1-100* cells, we introduced *pol12-216* (G325D) and *pol1-236* (D236N) alleles, which disrupt the interaction of Cdc13 and Stn1 with DNA polymerase α, respectively [40,41,43,85]. The *pol12-216* mutation was synthetically lethal with *stn1-ΔC*, as tetrad dissection of *STN1/stn1-ΔC POL12/pol12-216* diploids did not yield viable *stn1-ΔC pol12-216* spores (S4 Fig) [43]. Upon release from G1 into HU, *pol12-216 pol1-236* cells showed increased ssDNA at stalled replication forks, and ssDNA rose further

in *mec1-100 pol12-216 pol1-236* cells (Fig 7A). Since these alleles selectively weaken CST-Polα contacts, this finding suggests that Stn1 limits ssDNA, at least in part, by promoting fill-in synthesis via the Polα-primase complex.

Notably, introducing *stn1-L60F* into *mec1-100 pol12-216 pol1-236* cells reduced both the amount and extent of ssDNA to levels comparable to *pol12-216 pol1-236* cells, indicating that suppression by *stn1-L60F* is maintained despite weakened CST-Polα contacts (Fig 7A). Consistently, *mec1-100 stn1-L60F pol12-216 pol1-236* mutant displayed HU sensitivity similar to *mec1-100 stn1-L60F* (Fig 7B). Thus, while Stn1 can restrain ssDNA through Polα-primase-dependent fill-in, the suppression conferred by *stn1-L60F* does not require CST-Polα interface and is likely mediated primarily by an additional mechanism.

We therefore examined whether *stn1-L60F* limits ssDNA by reducing Mre11, Exo1, and Sgs1 association at stalled forks. ChIP-qPCR in cells synchronously released from G1 into HU showed that binding of Mre11, Exo1, and Sgs1 at ARS305 and ARS607 was increased in *mec1-100* relative to wild type, whereas this association was reduced in *mec1-100 stn1-L60F* cells (Fig 8). Abundance of these proteins was similar across strains (S5 Fig).

Taken together, these findings support a model in which Stn1 limits ssDNA accumulation in *mec1-100* both by pro-motimg fill-in synthesis by Polα-primase and by restricting the association of Mre11, Exo1, and Sgs1 with stalled forks, with the latter mechanism predominating in the suppression exerted by Stn1$^{L60F}$.

### Stn1 binds DNA DSBs and modulates both resection and checkpoint activation

It was recently reported that Stn1 limits the extent of ssDNA at a Cas9-induced DSB in budding yeast [65]. To further investigate the role of Stn1 in constraining ssDNA at DNA DSBs, we used derivatives of strain JKM139 in which a single irreparable DSB is generated at the *MAT* locus upon galactose-driven expression of the HO endonuclease gene. Because the homologous donor sequences *HML* and *HMR* are deleted, the break cannot be repaired by homologous recombination. ChIP-qPCR showed that Stn1 associates with the HO-induced DSB, and binding was increased in the presence of the *stn1-L60F* mutation (Fig 9A).

Consistent with results obtained after HU or phleomycin treatment (Fig 1E and 1G), HO induction led to reduced levels of phosphorylated, slower-migrating Rad53 in *stn1-L60F* cells compared with wild type, whereas Rad53 phosphorylation appeared earlier in *stn1-ΔC* cells (Fig 9B). Because Rad53 activation depends on ssDNA accumulation, these results suggest that Stn1 also counteracts ssDNA formation at DSBs.

We quantified ssDNA near the HO cut using a qPCR assay based on resistance to cleavage by restriction enzymes. If resection extends beyond a given restriction site, the DNA becomes resistant to cleavage and the fragment can be amplified by PCR with flanking primers. In line with reduced Rad53 phosphorylation, *stn1-L60F* cells showed decreased ssDNA at the HO-induced DSB relative to wild type, whereas ssDNA increased in *stn1-ΔC* cells that showed earlier Rad53 activation (Fig 9C).

Because Stn1 was shown to limit ssDNA at a Cas9-induced DSB by promoting fill-in synthesis via its interaction with the Polα-primase complex [65], we asked whether Stn1 also restrains the resection activities of Mre11, Exo1 and Sgs1 by limiting their association to the HO-induced DSB, as observed at stalled replication forks. ChIP-qPCR analysis revealed that the association of Mre11, Exo1, and Sgs1 at the HO-induced DSB was reduced in *stn1-L60F* cells compared to wild type, whereas it was increased in *stn1-ΔC* cells (Fig 9D). These results indicate that Stn1 limits ssDNA accumulation at DSBs not only by promoting fill-in synthesis but also by modulating the recruitment of Mre11, Exo1, and Sgs1.

## Discussion

The relationship between nascent DNA processing and replication-fork recovery remains incompletely understood. While a limited amount of resection can be required to generate substrates for homology-dependent restart and other remodeling reactions, excessive degradation of nascent DNA compromises fork integrity and promotes genome instability [86]. Abnormally long stretches of ssDNA are observed in *rad53* and *mec1* mutants exposed to HU [2,14], underscoring the role of checkpoint kinases in minimizing fork-proximal ssDNA and preserving the capacity for faithful recovery.

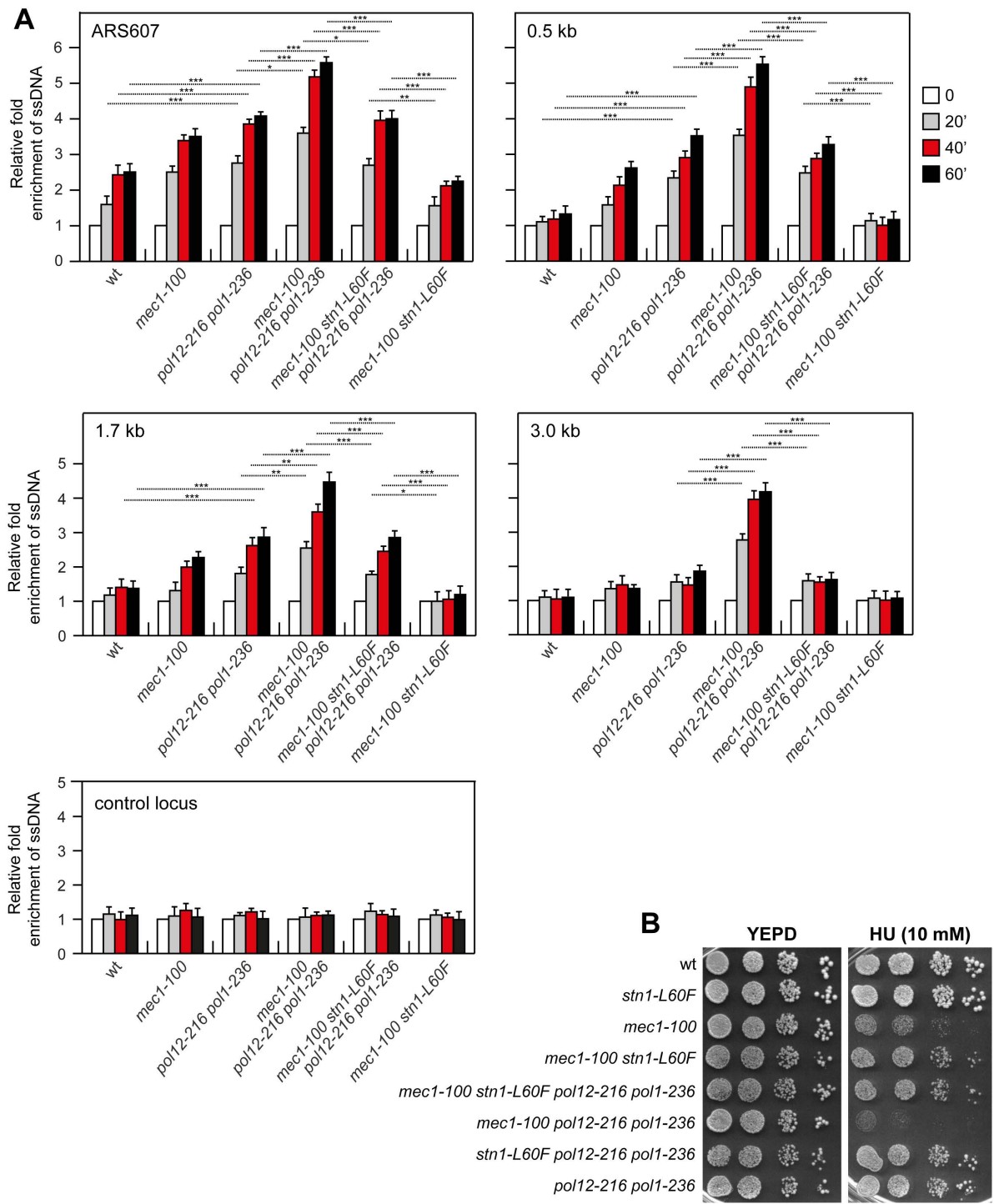

**Fig 7. Effect of *pol12-216* and *pol1-236* on the HU sensitivity and ssDNA generation of HU-treated *mec1-100* cells. (A)** Exponentially growing YEPD cell cultures were arrested in G1 with α-factor (time zero) and then released into YEPD containing 0.2M HU. ssDNA at different distances from ARS607 was assessed as described in Fig 2. ***$p < 0.005$, **$p < 0.01$, *$p < 0.05$ (Student's *t*-test). **(B)** Exponentially growing cell cultures were serially diluted (1:10) and each dilution was spotted out onto YEPD plates with or without HU.

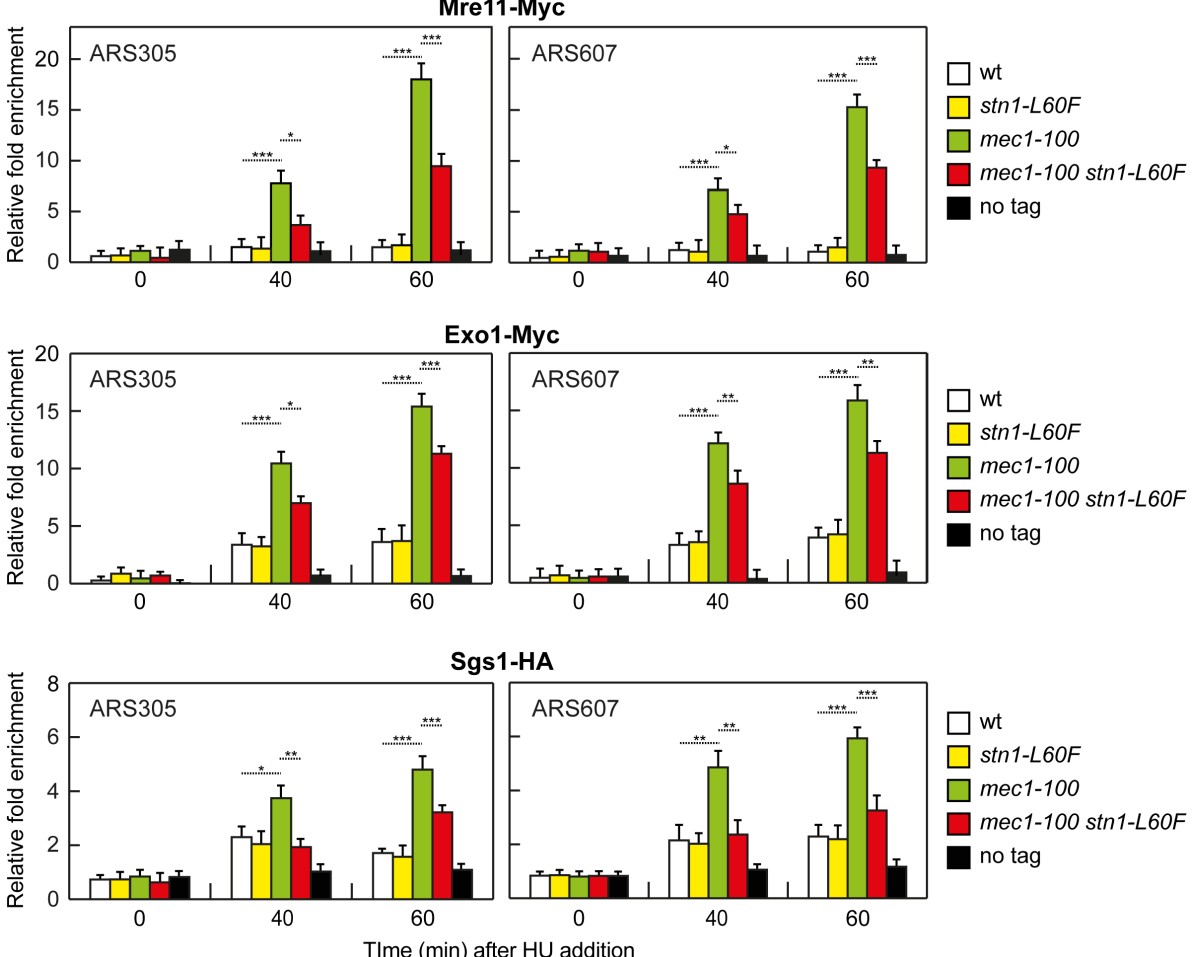

**Fig 8. Mre11, Exo1 and Sgs1 association at ARS305 and ARS607.** Cells were arrested in G1 with α-factor and then released into YEPD containing 0.2M HU at time zero. Relative fold-enrichment of Myc-tagged Mre11, Myc-tagged Exo1 and HA-tagged Sgs1 at ARS305 and ARS607 replication origins was determined after ChIP with an anti-Myc or an anti-HA antibody and qPCR analysis. Plotted values are the mean values ± s.d. from three independent experiments. ***$p < 0.005$, **$p < 0.01$, *$p < 0.05$ (Student's $t$-test).

In both yeast and mammals, the CST complex is best known for coordinating the transition from G-strand elongation by telomerase to C-strand synthesis by Polα-primase at telomeres [39–46,87–92]. Regulation of Polα-primase is not restricted to telomeres, as *S. pombe* CST also facilitates replication of repetitive genomic regions, including telomere-proximal subtelomeric regions and rDNA loci [56,57].

Here, we uncover an additional function for the CST subunit Stn1 in protecting replication forks from degradation when Mec1 activity is compromised. The gain-of-function *stn1-L60F* allele suppresses the HU sensitivity of *mec1-100* cells without restoring Rad53 phosphorylation, indicating that improved survival does not stem from checkpoint reactivation. Instead, suppression correlates with reduced ssDNA at stalled replication forks in *mec1-100 stn1-L60F*. Conversely, removing the C-terminal portion of Stn1 (*stn1-ΔC*), which impairs CST function at telomeres without affecting viability, has the opposite effect, increasing ssDNA and HU sensitivity of *mec1-100* cells. This ssDNA accumulation resembles that seen upon loss of Cdc13 function, either through temperature-sensitive mutations or degron-mediated degradation, which results in elevated levels of ssDNA at telomeres [47,93–95]. The protective function of Stn1 extends beyond stalled

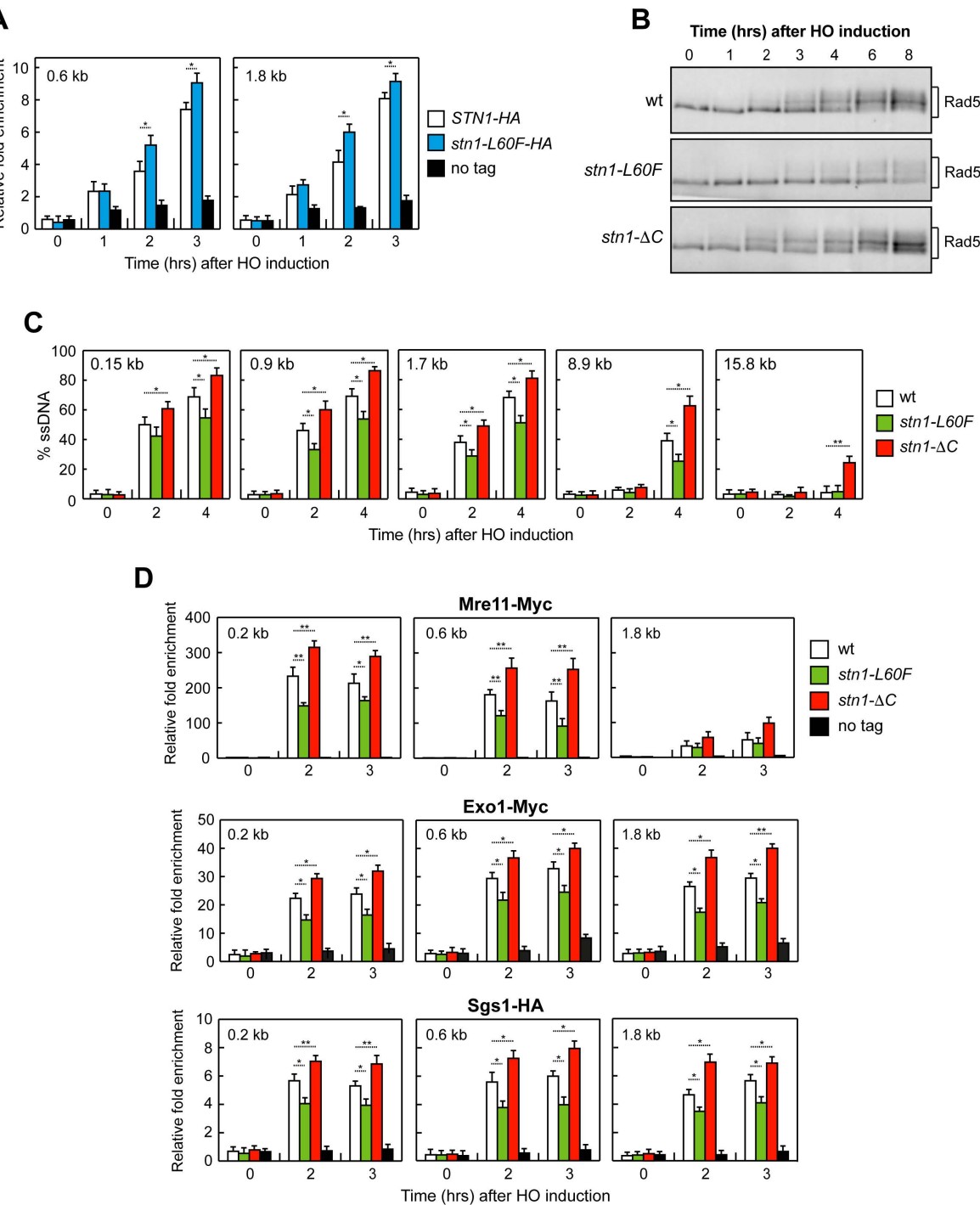

**Fig 9. Opposite effects of *stn1-L60F* and *stn1-ΔC* on checkpoint activation and ssDNA generation at DNA DSBs. (A)** Exponentially growing YEPR cell cultures of JKM139 derivative strains were transferred to YEPRG, followed by Stn1-HA and Stn1-L60F-HA ChIP at the indicated distances from the HO-cut site compared to untagged Stn1 (no tag). Data are expressed as fold-enrichment at the HO-cut site over that at a non-cleavable locus (*ARO1*), after normalization to the corresponding input for each time point. Fold-enrichment was then normalized to cut efficiency. Plotted values are the mean values ± s.d. from three independent experiments. \**p* < 0.05 (Student's *t*-test). **(B)** YEPR exponentially growing cell cultures of JKM139 derivative strains were transferred to YEPRG at time zero. Western blot analysis with an anti-Rad53 antibody of protein extracts from samples taken at the indicated times after HO induction. **(C)** Quantification of ssDNA by qPCR at the indicated distances from the HO cut site. Plotted values are the mean

values ± s.d. from three independent experiments. **$p < 0.01$, *$p < 0.05$ (Student's $t$-test). **(D)** Exponentially growing YEPR cell cultures of JKM139 derivative strains were transferred to YEPRG to induce HO expression. Relative fold-enrichment of Mre11-Myc, Exo1-Myc and Sgs1-HA at the HO-induced DSB was evaluated after ChIP with anti-Myc or anti-HA antibody and qPCR. Plotted values are the mean values ± s.d. from three independent experiments. **$p < 0.01$, *$p < 0.05$ (Student's $t$-test).

replication forks, as Stn1 limits ssDNA formation and checkpoint activation at DNA DSBs, supporting its role as a general antagonist of DNA resection.

Stn1 limits ssDNA formation at stalled forks by antagonizing Mre11, Exo1, and Sgs1, particularly when the checkpoint is impaired. Deletion of *EXO1* or *SGS1*, as well as expression of the nuclease-deficient *mre11-H125N* allele, reduces ssDNA in *mec1-100 stn1-ΔC* cells and prevents *stn1-ΔC* from further increasing HU sensitivity, indicating that the deleterious *stn1-ΔC* phenotype is mediated by these nucleases. Likewise, *stn1-L60F* fails to suppress the HU sensitivity of *mec1-100 mre11-H125N*, *mec1-100 exo1Δ*, and *mec1-100 sgs1Δ* cells. Interestingly, while *exo1Δ*, *sgs1Δ*, or *mre11-H125N* reduces ssDNA at stalled forks, each mutation increases the HU sensitivity of *mec1-100*. Thus, although excessive resection is detrimental, complete loss of any one pathway also worsens survival, implying that a basal, controlled level of processing is beneficial under nucleotide depletion. In this view, *stn1-L60F* does not abolish nuclease function; rather, it curbs excessive resection while preserving the basal processing needed for productive fork remodeling and restart, thus explaining why *stn1-L60F* no longer suppresses in *exo1Δ*, *sgs1Δ*, and *mre11-H125N* backgrounds.

Interestingly, in the processing of DNA DSBs, removing Sgs1-Dna2 or Exo1 results in minor defects, whereas inactivation of both leads to a severe resection defect, indicating that they control two partially overlapping pathways [78,79]. By contrast, at stalled replication forks, mutations in either Sgs1 or Exo1 mask the effects of Stn1-L60F and Stn1-ΔC, indicating that both factors, together with Mre11, participate in processing stalled replication forks. A plausible explanation is that DSB ends are relatively simple substrates for 5'-3' resection, so either Sgs1-Dna2 or Exo1 can perform the reaction. In contrast, a stalled fork is a composite substrate that can contain reversed arms, RPA-coated ssDNA gaps, and incomplete Okazaki fragments on the lagging strand, creating different structures that make both Exo1 and Sgs1-Dna2 individually important for fork processing but functionally redundant at DSBs.

Work in yeast and human cells has shown that CST limits ssDNA at DSBs via post-resection Polα-primase-dependent fill-in, protects nascent DNA at stalled forks by reducing MRE11 association, and directly inhibits EXO1 and BLM-DNA2 in vitro [61–67]. Our results in budding yeast support both facets: weakening CST-Polα contacts with *pol12-216* and *pol1-236* increases ssDNA at stalled forks, consistent with a contribution from Polα-primase to limiting ssDNA via fill-in. This behavior mirrors what was observed at DSBs, where weakening CST-Polα contacts with *pol12-216* and *pol1-236* decreases ssDNA generation [65]. Furthermore, in *mec1-100* cells the L60F mutation reduces Mre11, Exo1, and Sgs1 association at ARS305 and ARS607 relative to wild type. Loss of fill-in capacity caused by the *pol12-216* and *pol1-236* mutations does not impair the ability of *stn1-L60F* to suppress, indicating that *stn1-L60F* acts primarily by limiting nuclease association at stalled forks rather than by converting ssDNA back to dsDNA. Stn1<sup>L60F</sup> displays higher occupancy at early origins upon HU and stronger binding at a DSB than wild type. AlphaFold-based modeling predicts that the Leu-Phe substitution enhances contacts within ssDNA and the CST interface. We therefore propose that increased ssDNA binding by Stn1<sup>L60F</sup> primarily acts as a shield, reducing access of Mre11, Exo1, and Sgs1 by steric/capture competition. This enhanced occupancy does not by itself boost fill-in synthesis, which depends on CST interactions with Polα.

Whether this limitation in ssDNA formation at stalled forks by antagonizing Mre11, Exo1, and Sgs1 is specific for Stn1 or involves the whole CST complex remains to be determined. Bioinformatic simulations suggest that the higher apparent affinity for ssDNA in the mutant CS<sup>L60F</sup>T complex reflects contributions from both Stn1 and Cdc13 contacting DNA, with the phenylalanine in place of leucine allowing a more stable interface. This points to a protective role of the entire CST complex, not Stn1 alone.

We propose that, under nucleotide depletion, *mec1-100* permits continued DNA unwinding and accumulation of incomplete Okazaki fragments, which deplete replication factors and expose nascent DNA to degradation by Mre11, Exo1, and Sgs1-Dna2 [28,29]. Stn1 promotes fork restart by facilitating Polα-primase-dependent fill-in synthesis that shortens gaps and by limiting nuclease association at fork-proximal ssDNA. *stn1-L60F* enhances the ssDNA-binding/shielding aspects of Stn1, restoring a balance in which basal processing is preserved but excessive resection is prevented (Fig 10).

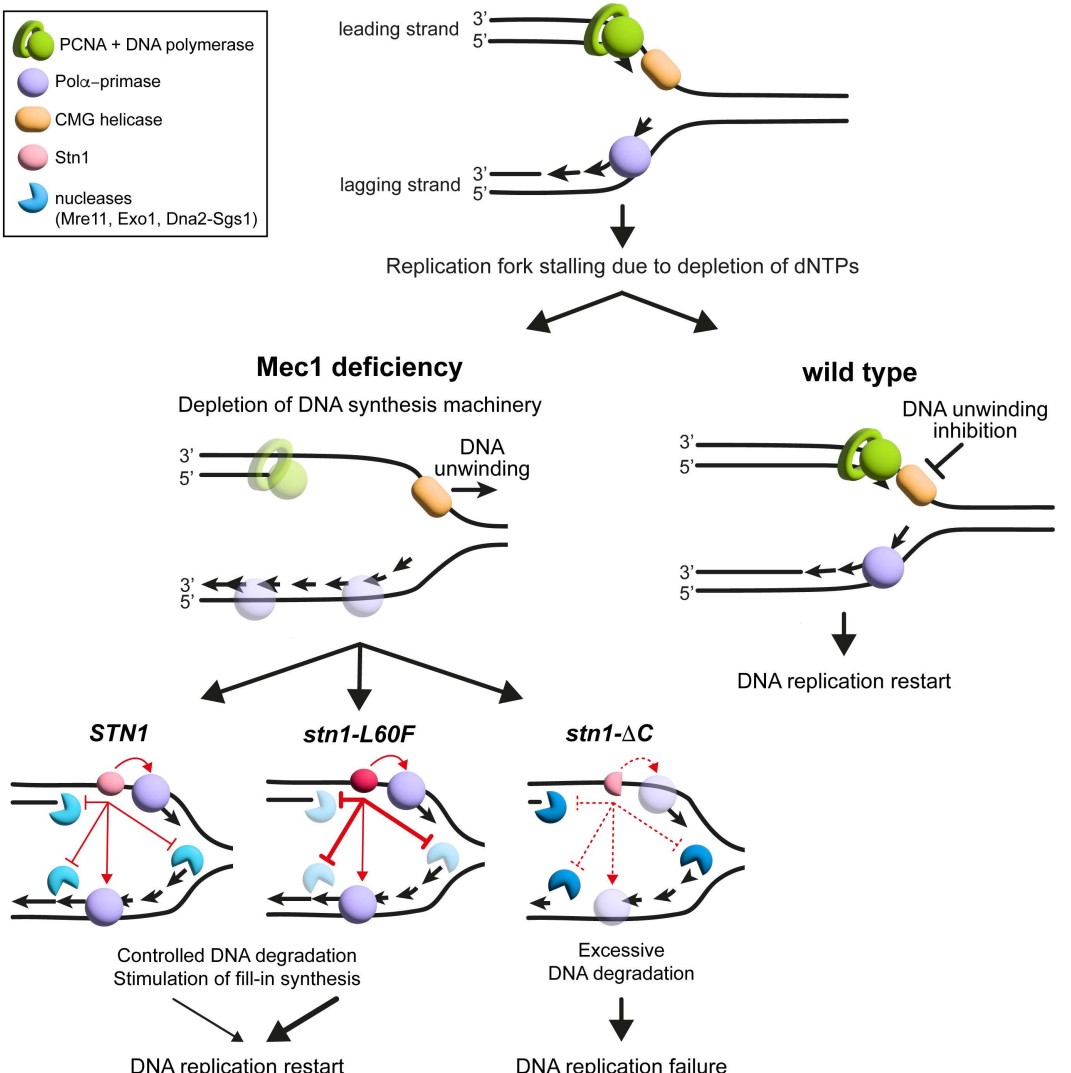

**Fig 10. Model of Stn1 role in supporting Mec1 function at stalled replication forks.** dNTP depletion stalls replication forks. When Mec1 is compromised (*mec1-100*), the CMG helicase keeps unwinding, leading to excessive accumulation of incomplete Okazaki fragments, depletion of replication factors, aberrant fork restart, and exposure of DNA to nuclease attack. When Stn1 is present, Polα-primase-mediated fill-in synthesis is promoted and the fork-proximal recruitment/retention of Mre11, Exo1, and Dna2-Sgs1 on DNA is restrained, enabling restart. The gain-of-function *stn1-L60F* mutation increases Stn1 association with stalled replication fork, thereby limiting the association of Mre11, Exo1, and Dna2-Sgs1 and the resulting ssDNA generation more effectively than wild-type Stn1. In contrast, the loss-of-function *stn1-ΔC* mutation fails to restrain the resection activity of Mre11, Exo1, and Dna2, leading to extensive DNA degradation and replication failure. A limited nuclease activity remains beneficial for fork remodeling in *mec1-100*. In wild-type cells, checkpoint signaling restrains CMG-driven unwinding, prevents excessive Okazaki-fragment formation, preserves replication factors, and permits proper DNA replication restart.

Interestingly, human CST is phosphorylated in an ATR-dependent manner [96], suggesting that checkpoint signaling may regulate CST activity in response to DNA damage or replication stress.

In sum, our data identify Stn1 as a key modulator that protects both stalled replication forks and DSBs from excessive nucleolytic processing. This function becomes particularly important when Mec1 activity is compromised, revealing a functional interplay between CST and the replication checkpoint in preserving genome stability under replication stress.

## Materials and methods

### Yeast strains and media

*Saccharomyces cerevisiae* is the experimental model used in this study. Strain genotypes are listed in S2 Table. Strain JKM139, used to detect DSB resection, was kindly provided by J. Haber (Brandeis University, Waltham, USA). The *pol12-216* mutant was kindly provided by D. Shore (Université de Genève, Geneva, Switzerland). The *pol1-236* mutant was kindly provided by V.A. Zakian (Princeton University, New Jersey, USA). Gene disruptions and tag fusions were constructed by one-step PCR homology cassette amplification and standard yeast transformation methods. Cells were grown in YEP medium (1% yeast extract, 2% bactopeptone) supplemented with 2% glucose (YEPD), 2% raffinose (YEPR) or 2% raffinose and 3% galactose (YEPRG). All the experiments have been performed at 25°C.

### Search for suppressors of the HU sensitivity of *mec1-100* cells

To search for suppressor mutations of the HU sensitivity of *mec1-100* cells, $5 \times 10^6$ *mec1-100* cells were plated on YEPD in the presence of HU. Survivors were crossed to wild-type cells to identify by tetrad analysis the suppression events that were due to single-gene mutations. Genomic DNA from single-gene suppressors was analyzed by next-generation Illumina sequencing (IGA technology services). To confirm that the *stn1-L60F* allele was responsible for the suppression, the *URA3* marker gene was integrated downstream of the *stn1-L60F* stop codon and the resulting strain was crossed to wild-type cells to verify by tetrad dissection that the suppression of the *stn1-L60F* sensitivity co-segregated with the *URA3* marker gene.

### Spot assays

Cells grown overnight were diluted to $1 \times 10^7$ cells/mL. 10-fold serial dilutions were spotted on YEPD with or without the indicated concentration of HU. Plates were incubated for 3 days at 25°C.

### Southern blot analysis of telomere length

To determine the length of native telomeres, XhoI-digested genomic DNA was subjected to 0.8% agarose gel electrophoresis and hybridized with a $^{32}$P-labeled poly(GT) probe. Standard hybridization conditions were used.

### Protein extract preparation and western blotting

Protein extracts for western blot analysis were prepared by trichloroacetic acid (TCA) precipitation, as previously described [97]. Supernatants containing the solubilized proteins were separated on 10% polyacrylamide gels. Rad53 was detected by using an anti-Rad53 polyclonal antibody (ab104232) (1:2000) from Abcam. HA-tagged proteins were detected by using an in-house anti-HA (12CA5) (1:2000) antibody. Myc-tagged proteins were detected by using anti-Myc (Ab32) (1:2000) antibody from Abcam. Images were collected using the ChemiDoc (Bio-Rad) and ImageLab software.

### Quantification of ssDNA by qPCR at DNA replication forks

ssDNA was quantified as previously described [74]. Genomic DNA was digested with the restriction enzyme SspI, with a parallel mock reaction lacking the enzyme. Digested and mock-digested DNA samples were amplified by qPCR using Sso-Fast EvaGreen Supermix (Bio-Rad) and primer pairs flanking the SspI site. Reactions were run on a Bio-Rad CFX Connect

Real-Time System and analyzed with CFX Maestro 1.1. A control locus located 20 kb from ARS1103 and 27 kb from ARS1102 on chromosome XI was used for normalization. Oligonucleotide used for ssDNA qPCR are listed in S3 Table.

## Quantification of ssDNA by qPCR at a DNA DSB

qPCR analysis of resection at the *MAT* locus was performed as previously described [98]. Genomic DNA was digested with the restriction enzymes SspI and RsaI, with a parallel mock reaction lacking the enzymes. qPCR was performed on both digested and mock-digested samples using SsoFast EvaGreen Supermix (Bio-Rad) and primer pairs flanking the SspI or RsaI sites. Reactions were run on a Bio-Rad CFX Connect Real-Time System and analyzed with CFX Maestro 1.1. For each time point, Ct values were first normalized to the corresponding mock sample and then to an amplicon within the *KCC4* control gene. Values were further adjusted for HO cutting efficiency, measured by qPCR with primers flanking the HO recognition site. The percentage of HO cutting was calculated by comparing Ct values in undigested samples before and after HO induction. Oligonucleotides used for ssDNA qPCR are listed in S3 Table.

## Chromatin immunoprecipitation and qPCR

ChIP was performed as previously described [99]. Wash buffers were: SDS buffer (0.025% SDS, 50 mM HEPES pH 7.5, 140 mM NaCl, 1 mM EDTA, 1%), High-Salt buffer (1 M NaCl, 50 mM HEPES pH 7.5, 1 mM EDTA), and T/L buffer (200 mM Tris pH 8.0, 250 mM LiCl, 0.5% sodium deoxycholate, 1 mM EDTA, 0.15% IGEPAL). For Stn1 recruitment at DNA replication origins, minor modifications were introduced. Specifically, after immunoprecipitation, samples were washed with Lysis buffer (50 mM HEPES pH 7.5, 140 mM NaCl, 1 mM EDTA, 1% Triton X-100, 0.1% sodium deoxycholate, protease inhibitors) and Wash buffer (100 mM Tris pH 8.0, 250 mM LiCl, 0.5% sodium deoxycholate, 1 mM EDTA, 1% Triton X-100). Exo1-Myc and Mre11-Myc were immunoprecipitated with anti-Myc (Ab32; Abcam) at 1:2000. Sgs1-HA and Stn1-HA were immunoprecipitated with an in-house anti-HA antibody (12CA5). For ChIP at replication forks, data are expressed as fold-enrichment at ARS607 and ARS305 relative to a control region 14 kb from ARS607, after normalizing each ChIP signal to the corresponding input at each time point. For ChIP at the *MAT* locus, data are expressed as fold-enrichment at the HO-induced DSB relative to the non-cleaved *ARO1* locus, after input normalization at each time point. Fold-enrichment was further normalized to the DSB induction efficiency, assessed by Southern blot analysis. Oligonucleotides used for ChIP-qPCR are listed in S3 Table.

## Bioinformatic prediction and analysis

A model of the budding yeast wild-type CST complex was generated using the AlphaFold 3 server [100] (model quality assessment for the protein components is shown in S3 Fig) by submitting the Cdc13, Stn1, and Ten1 sequences from the UniProtKB database (https://www.uniprot.org/) together with a 20-nt ssDNA oligonucleotide matching the typical yeast telomeric G-tail repeat as in [84] (5'-GGGTGTGTGGTGGGTGTGGT-3'), to optimize affinity for the CST complex. The L60F mutation was introduced in the Stn1 structure using the PyMOL mutagenesis tool. Both wild-type and mutant models were relaxed with the HADDOCK 2.4 server water-refinement protocol [101] (https://wenmr.science.uu.nl/haddock2.4/settings#refinement) under standard parameters: a series of short MD simulations with explicit solvent with position restraint on the α backbone of the protein, thus allowing the polypeptide chains to adjust; then 1,250 MD steps at 300 K with positional restraints on residues not involved in intermolecular contacts within 5 Å; finally, the temperature is progressively lowered (1,000 MD steps at 300, 200, and 100 K) with positional restraints on heavy atoms of the complex, excluding interface atoms. The protocol produced four models for each system and interface/binding energy evaluations (energy of the complex relative to the individual components). Structural visualization was performed with PyMOL and UCSF Chimera; hydrogen-bond analyses were carried out in UCSF Chimera. Domain architecture schematics for CST components were generated with ProToDeviseR (https://matrinet.shinyapps.io/ProToDeviseR/) using domain annotations from CDD (https://www.ncbi.nlm.nih.gov/cdd) and InterPro (www.ebi.ac.uk/interpro/).

## Quantification and statistical analysis

Data are expressed as mean values ± standard deviation. Statistical analyses were performed using Microsoft Excel Professional 365 software. *P* values were determined by using an unpaired two-tailed *t*-test. No statistical methods or criteria were used to estimate sample size or to include or exclude samples.

## Supporting information

**S1 Table. HADDOCK 2.4 binding-energy evaluation for water-refined models of the CST wild-type and CS^L60F^T mutant complexes.**
(DOCX)

**S2 Table. *S. cerevisiae* strains used in this study.**
(DOCX)

**S3 Table. Oligonucleotides used in this study.**
(DOCX)

**S1 Fig. Effect of *stn1-L60F* and *stn1-ΔC* on telomere length and on the HU sensitivity of *mec1Δ sml1Δ* cells. (A)** Exponentially growing cell cultures were serially diluted (1:10) and each dilution was spotted out onto YEPD plates with or without HU. **(B)** XhoI-cut genomic DNA from exponentially growing cells was subjected to Southern blot analysis using a radiolabeled poly(GT) telomere-specific probe.
(TIF)

**S2 Fig. ssDNA in *mec1-100* depends on Mre11, Exo1 and Sgs1.** Exponentially growing YEPD cell cultures were arrested in G1 with α-factor (time zero) and then released into YEPD containing 0.2M HU. Genomic DNA prepared at different time points after α-factor release was either digested or mock-digested with SspI and used as a template in qPCR. The value of SspI-digested over non-digested DNAs was determined for each time point after normalization to an amplicon on chromosome XI that does not contain SspI sites. The data shown are expressed as fold-enrichment in ssDNA at different time points after α-factor release in HU relative to the α-factor (time zero) (set to 1.0). A locus containing SspI sites on chromosome XI is used as a control (control locus). Plotted values are the mean values ± s.d. from three independent experiments. \*\*\**p* < 0.005, \*\**p* < 0.01, \**p* < 0.05 (Student's *t*-test).
(TIF)

**S3 Fig. Quality assessment for AlphaFold 3-generated model of the budding yeast CST complex bound to a telomeric ssDNA tail. (A)** The model generated by AlphaFold 3 for the budding yeast Cdc13-Stn1-Ten1 complex bound to a 20-nt ssDNA is represented as a cartoon and coloured according to the lDDT score. **(B)** Expected Position Error calculated by AlphaFold 3 predictor is represented for the residues of Cdc13, Stn1, and Ten1, numbered consecutively.
(TIF)

**S4 Fig. Synthetic lethality between *pol12-216* and *stn1-ΔC* alleles.** Meiotic tetrads were dissected on YEPD plates that were incubated at 25°C, followed by spore genotyping.
(TIF)

**S5 Fig. Mre11, Exo1 and Sgs1 protein levels.** Western blot with an anti-HA or an anti-Myc antibody of extracts used for the ChIP analysis shown in Fig 8. The same amount of extracts was separated by SDS-PAGE and stained with Coomassie Blue as a loading control.
(TIF)

**S1 Data. Numerical data used for the generation of graphs.**
(XLSX)

**S2 Data. Uncropped western blots.**
(PDF)

## Acknowledgments

We thank J. Haber, D. Shore and V.A Zakian for providing yeast strains and M. Clerici for critical reading of the manuscript.

## Author contributions

**Conceptualization:** Erika Casari, Flavio Corallo, Maria Pia Longhese.

**Data curation:** Erika Casari, Flavio Corallo, Luca Edoardo Milani, Renata Tisi, Maria Pia Longhese.

**Formal analysis:** Erika Casari, Flavio Corallo, Renata Tisi, Maria Pia Longhese.

**Funding acquisition:** Maria Pia Longhese.

**Investigation:** Erika Casari, Flavio Corallo, Luca Edoardo Milani.

**Methodology:** Erika Casari, Flavio Corallo, Luca Edoardo Milani, Renata Tisi, Maria Pia Longhese.

**Project administration:** Maria Pia Longhese.

**Resources:** Maria Pia Longhese.

**Supervision:** Maria Pia Longhese.

**Validation:** Maria Pia Longhese.

**Visualization:** Erika Casari, Flavio Corallo, Renata Tisi, Maria Pia Longhese.

**Writing – original draft:** Maria Pia Longhese.

**Writing – review & editing:** Erika Casari, Flavio Corallo, Renata Tisi, Maria Pia Longhese.

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
