## [Decision Letter · Decision Letter 0]

28 Jul 2025

PGENETICS-D-25-00691

Stn1 supports Mec1 function in protecting stalled replication forks from degradation

PLOS Genetics

Dear Dr. Longhese,

Thank you for submitting your manuscript to PLOS Genetics. After careful consideration, we feel that it has merit but does not fully meet PLOS Genetics's publication criteria as it currently stands. Therefore, we invite you to submit a revised version of the manuscript that addresses the points raised during the review process.

Please submit your revised manuscript within 60 days Sep 26 2025 11:59PM. If you will need more time than this to complete your revisions, please reply to this message or contact the journal office at plosgenetics@plos.org. Please include the following items when submitting your revised manuscript:

We look forward to receiving your revised manuscript.

Kind regards,

Ashok Bhagwat, Ph.D.

Academic Editor

PLOS Genetics

Hongbin Ji

Section Editor

PLOS Genetics

Aimée Dudley

Editor-in-Chief

PLOS Genetics

Anne Goriely

Editor-in-Chief

PLOS Genetics

**Additional Editor Comments:**

While all the reviewers found the results presented here to be interesting, there was different levels of enthusiasm about its significance and novelty. It was felt that the manuscript did not adequately point out significance of relevant literature in both yeast and mammalian cells. There were also serious concerns about the lack of an overall model which is consistent with the existing literature and the data presented here.

**Journal Requirements:**

At this stage, the following Authors/Authors require contributions: Erika Casari, Flavio Corallo, Luca Edoardo Milani, and Maria Pia Longhese. Please ensure that the full contributions of each author are acknowledged in the "Add/Edit/Remove Authors" section of our submission form.

The list of CRediT author contributions may be found here: https://journals.plos.org/plosgenetics/s/authorship#loc-author-contributions

4) We note that your Data Availability Statement is currently as follows: "All relevant data are within the manuscript and its Supporting Information files.". Please confirm at this time whether or not your submission contains all raw data required to replicate the results of your study. Authors must share the “minimal data set” for their submission. PLOS defines the minimal data set to consist of the data required to replicate all study findings reported in the article, as well as related metadata and methods (https://journals.plos.org/plosone/s/data-availability#loc-minimal-data-set-definition).

**Reviewers' comments:**

Reviewer's Responses to Questions

**Comments to the Authors:**

Reviewer #1: PGenetics D 25-00691

Stn1 supports Mec1 function in protecting stalled replication forks from degradation.

This is an excellent study of one feature of replication fork biology in budding yeast. The authors design strategies to identify proteins important in fork stability, identified because such proteins restore cell viability in cells with low levels of dNTP (conferring hydroxyurea (HU) sensitivity) in a mutant-sensitized cell (mec1-100 mutation). In brief, they identify a role for the protein Stn1protein that is a part of a heterotrimeric complex called CST (Cdc13, Stn1, Ten1) that is known to act at telomeres, and now at internals sites as well. CST that binds ssDNA at telomeres to facilitate telomere biology. They identify suppressors of mec1-100 HUs, one of which is Stn1-L60F. They characterize Stn1-L60F, along with a Stn1�C mutation, and mutations in other proteins known to have roles in DNA replication fork biology (exo1, sgs1, mre11-125, pol2-216).

The data involve growth assays, assays of ssDNA by a qPCR strategy, ChIP of proteins to stalled forks (from a specific origin, ARS607), and at a DSB (HO induced), and westerns of Rad53 checkpoint protein kinase. The data are impeccable, as per usual from the Longhese lab.

Comments:

1. I did have a hard time tracking a certain part of their narrative: The paper would benefit from a model!!! I think a model might be a challenge to draw up for one feature of their data that confused me. Generally, Stu1-L60F binds ssDNA, prevents Sgs1, Exo1 and Mre11 from binding, and therefore prevents these nuclease/nuclease-associated proteins from converting dsDNA to ssDNA. All this in the context of the mec1-100 hypomorphic mutation. Fair enough.

What I did not understand is the data in Figure 2, 3 and Fig4. First off, they don't show ssDNA data for mec1-100 sgs1, mec1-100 exo1, mecI-100 mre11-H125N (I think they didn't--they should explain why they don't). So I infer that formation of ssDNA requires these three proteins.

Then, my confusion. Just looking at Figure 4 and Fig 6 for Stn1L60F: In a mec1-100 mutant, Stn1L60F prevents formation of ssDNA and increases HU viability. But they wisely add to that double mutant a mutation in sgs1 or exo1 pr mre11-H125N (e.g. triple mutant mec1-100, Stn1L60F exo1), HU viability is low! Thus, the HU viability in mec1-100 Stn1 L60F requires intact Sgs1, Exo1, and Mre11 (as mre11-125)! Yet the model is that Stn1-L60F prevents/minimizes Sgs1, Exo1, Mre11 from binding ssDNA near stalled forks?

The authors should provide a plausible explanation for this data, and provide a model!!!

And, what about Stn1 in the CST complex (with Cdc13 and Ten1). They should at least speculate on if they think Stn1 is acting with Cdc13 and Ten1 at stalled forks, or is acting alone, or is unknown.

The paper for this reviewer is otherwise solid and extremely well done.

Reviewer #2: This is a fascinating paper, adding much to our understanding of the regulation of end-resection at telomeres, at stalled replication forks and at double-strand breaks. The role Stn1 (and presumably the rest of CST) is an unexpected complication in our understanding of end-processing in yeast. Apparently in contrast to recent conclusions concerning mammalian CST, where CST is viewed as filling-in the just-resected end, in yeast it appears that CST more directly impairs all of the several exonuclease activities.

An apparently important difference that emerges from this study is how Mre11, Exo1 and Sgs1 affect the formation of ssDNA in replication-arrested cells versus the creation of a double-strand break. In end-processing of a DSB, Sgs1 (with Dna2) and Exo1 are usually viewed as redundant and only in a double mutant is there a severe effect on long-range resection. The results presented here imply that at stalled replication forks, both Exo1 and Sgs1 are required, in addition to Mre11, as mutants of each mask both the effects of Stn1-L60F and Stn1-�C. This is surprising and confusing; the authors might provide more discussion of this point.

Specific comment

1) If Stn1 acts apparently upstream of all three exonucleases, how is it recruited to ssDNA/DSB regions? At telomeres, Yku70/80 is a prominent resident and it is recruited to DSBs quickly. Does Stn1 recruitment depend on Yku70/80?

2) Fig. 2. We don’t understand the plots of enrichment of ssDNA at ARSs. Shouldn’t these values be normalized to 1 at t = 0, as in the plots above in the same figure?

3) Another key point in the MS is that the effects of Stn1 mutants are independent of activating the DNA damage checkpoint, even though Rad53 activation is quite strongly affected. In that case, what are the phenotypes of the mutants with mec1 sml1?

4) Fig. 8 : the assays for ssDNA are not coincident with the positions of the ChIPs for the exonucleases. Why were different primers used? Are these from the same experiments, and the same side of the DSB?

5) It would be good to show the effect of Stn1 and the nucleases for some DSB repair event, as the effects on resection, though apparently almost all statistically significant, are surprisingly modest, whereas the effects on Rad53 phosphorylation are strikingly large. More generally, how to explain how these small differences have such large effects.

6) cdc13-1 strains arrest but then adapt (Toczyski et al. 1997). Does Stn1-L60F or �C affect this behavior? Also, if Stn1-�C exhibits the same end-protection defects as ts mutants of cdc13 (l. 308), in what way is Cdc13 more essential to telomere maintenance?

7) Is Stn1-�C dominant over stn1-L60F or vice versa? Does this relationship help suggest how many copies of the complex may act at any one site?

8) This last point is based on a recent preprint about yeast end-resection and actin: https://www.researchsquare.com/article/rs-5125308/v1

Given recent studies of how the Arp2/3 complex blocks resection in yeast, as it apparently does in mammalian cells, one can only wonder how those results are tied into the role of Stn1. The authors might want to comment on this.

9) the authors might also comment on the recently published paper: https://www.cell.com/action/showPdf?pii=S2666-979X%2825%2900203-4

Reviewer #3: In this study Casari et al, investigated the link between CST complex and checkpoint response in response to replication stress. With a genetic screen they identified a mutation of STN1 (stn1-L60F) that slightly restores viability of mec1-100 cells in response to HU. They showed that Stn1 limits the accumulation of ssDNA at replication forks, counteracting resection by Mre11, Exo1 and Sgs1/Dna2.

It would have been fair to state in the introduction that human CST complex was already known to protect stalled replication forks by directly blocking MRE11 degradation of nascent-strand DNA. On the top of that, several studies reported also that Stn1-Ten1 protect blocked forks by antagonising Exo1-extensive resection at DNA ends in fission yeast. Knowing that, the novelty of this study is less. This work rather brings some insight on the connection between Mec1-depedent checkpoint response and Stn1 which is interesting although not fully addressed. However, I have the impression that only the surface has been touched and that we reinvent the wheel in several aspects. The authors claimed that association of Stn1-L60F is increased at the fork however the demonstration by ChIP doesn’t look very convincing. I am not convinced either by the fact that role of protection by Stn1 is independent on the fill-in reaction. Along the same line, this work is also in direct contradiction with a recent study that shows that CST acts after resection at DSB (DOI: 10.1016/j.xgen.2025.100947).

Overall, although the discovery of this new allele of stn1 seems interesting, the novelty of the proposed mechanism is questioned and it is not clearly established how this mutation impacts the function of CST.

Specific points:

In figure 1, it will helpful to insert a scheme of the CST elements with functional domains and their own domain of interaction and interaction with polymerase alpha (Pol1 and Pol12), the telomerase, DNA,… L60 should lie in the Ten1 interaction domain. Alphafold modelling is recommended for this kind of analysis. How does this mutation impact interaction with Ten1 and Cdc13 and ssDNA binding? This was not assessed and discussed.

Figure 1A: The rescue of mec1-100 allele by stn1-L60F is mild. Survival curves with statistical analysis will be more appropriate to visualize this rescue and to convince the readers.

Figure 1E: I don’t see a fundamental change in mec1-100 vs mec1-100 stn1-L60F double. I would say epistatic unless quantifications of Western blot are shown. Thus, the statement that stn1-L60F impairs CP activation is not clearly demonstrated.

Figure 2B

The demonstration that stn1-L60F reduces ssDNA accumulation seems convincing, but the demonstration that Stn1-L60F accumulates at stalled forks is not. According to Figure 1F, the effect of Phleomycin on the delay of CP activation is stronger. Using Phleomycin could a way to exacerbate the ChIP signal.

Lane 219 p 9 : should be figure 4 ?

This statement p9 “Based on these findings, Stn1 may limit ssDNA generation at stalled replication forks by counteracting the resection activity of Mre11, Exo1, and Dna2-Sgs1 via polymerase α–primase-dependent fill-in synthesis and/or by directly inhibiting these nucleases ». The authors need to mention that CST is known to protect stalled replication forks by directly blocking MRE11 degradation of nascent-strand DNA in human cell lines and that reported that Stn1-Ten1 protect blocked forks by antagonising Exo1-extensive resection in fission yeast. This assumption has already been proposed and demonstrated by previous studies.

Figure 6 : I have a concern regarding the interpretation of the data with the pol12-216 allele. The authors states page 9 line 240 that this mutation G325D in Pol12 disrupts the interaction with Polymerase alpha and CST. According to Grossi et al., disruption is only partial (two-fold). In addition, it is likely that Cdc13 still interacts with Pol1 subunit pol12-216 mutant. Why the author did not combine with both pol12-216 with pol1-236 that abolishes interaction between Cdc13 and Pol1. Thus, although the interaction is reduced pol12-216 mutant, fill-in reaction can still occur.

Is possible that L60F mutation compensate the interaction with Pol12-G325D? This was not addressed. The combination of mec1-100 with pol12-216 mutant is synthetic lethal but we can’t rule out that this effect is not due only to a decrease in interaction with CST.

Figure 8: A recent publication demonstrates the role of CST in fill-in reaction at Cas9-induced DSB (PMID: 40675158). This work clearly overlaps with and contradicts the result of Figure 8. While the authors claim that CST protects from resection independently of fill-in synthesis this new study demonstrates that CST acts after resection to limit ssDNA accumulation by fill-in reaction.

Deletion of rif1 is known to restore viability of dysfunctional stn1 alleles by likely firing new origins. How deletion of rif1 would impact mec1-100 and stn1-deltaC alleles. Would it be epistatic or additive when combined to stn1-L60F?

**Have all data underlying the figures and results presented in the manuscript been provided?**

Reviewer #1: Yes

Reviewer #2: **No: **

Reviewer #3: Yes

PLOS authors have the option to publish the peer review history of their article (what does this mean?). If published, this will include your full peer review and any attached files.

Reviewer #1: **Yes: ** Ted Weinert

Reviewer #2: **Yes: ** Jim Haber

Reviewer #3: No

**Figure resubmission:**
---

## [Decision Letter · Decision Letter 1]

10 Oct 2025

Dear Dr Longhese,

We are pleased to inform you that your manuscript entitled "Stn1 supports Mec1 function in protecting stalled replication forks from degradation" has been editorially accepted for publication in PLOS Genetics. Congratulations!

Yours sincerely,

Ashok Bhagwat, Ph.D.

Academic Editor

PLOS Genetics

Hongbin Ji

Section Editor

PLOS Genetics

Aimée Dudley

Editor-in-Chief

PLOS Genetics

Anne Goriely

Editor-in-Chief

PLOS Genetics

BlueSky: @plos.bsky.social

Comments from the Editor:

All the reviewers believe that the manuscript is considerably improved and should be accepted. However, Reviewer 2 is still unclear about why different alleles of Stn1 show different phenotypes and suggests an experiment to clarify this issue. Please either include results of such an experiment in the final version of the manuscript or a clearer explanation of your results. Also, Reviewer #3 would like to see some relevant literature cited in the final version. Please do so in the relevant parts of the manuscript.

Reviewer's Responses to Questions

**Comments to the Authors:**

Reviewer #1: The submission is completely fine with me.

Reviewer #2: Two very different alleles of the Stn1 component of CST exhibit surprisingly different phenotypes. The authors now offer a model that isn't detailed enough to make clear how these phenotypes arise. It is for this reason that I strongly urge the authors to include the data on the heterozygous double mutant, which may help others sort out what is happening (reviewer 2, point 7). Also, was this done by creating a diploid as opposed to adding the alleles to a haploid? Were the diploids homozygous for each mutant also examined? I can imagine that viability in a diploid might be quite different.

Similarly, in discussing the effects of removing Mre11, Exo1 or Sgs1 activity (l. 222) the authors should point out that the effect of exo1 alone or sgs1 alone is distinctly different from their apparently redundant roles in resection broken chromosome ends. The big limitation is that, since most of the effects seen here are in the context of a poorly understood mec1-100 mutation, which is clearly different from the absence of Mec1, it is hard to pinpoint what is happening.

Reviewer #3: The manuscript is improved. I insist of the fact that in introduction (lane 95), in result section (lane 275) and in discussion (lane 342), the authors should also acknowledge Ishikawa’s Lab work (NAR 2021, 2016). Most importantly, it should be clearly stated there that Stn1 is important to maintain replication of rDNA, subtelomeres and telomeres acting at the fork to limit ssDNA accumulation, indicating that Stn1 acts broadly (not only at telomere region) to maintain fork stability.

**Have all data underlying the figures and results presented in the manuscript been provided?**

Reviewer #1: None

Reviewer #2: Yes

Reviewer #3: Yes

PLOS authors have the option to publish the peer review history of their article (what does this mean?). If published, this will include your full peer review and any attached files.

Reviewer #1: No

Reviewer #2: No

Reviewer #3: No

**Data Deposition**

http://datadryad.org/submit?journalID=pgenetics&manu=PGENETICS-D-25-00691R1

**Press Queries**

---

## [Editor Report · Acceptance letter]

PGENETICS-D-25-00691R1

Stn1 supports Mec1 function in protecting stalled replication forks from degradation

Dear Dr Longhese,

We are pleased to inform you that your manuscript entitled "Stn1 supports Mec1 function in protecting stalled replication forks from degradation" has been formally accepted for publication in PLOS Genetics! Your manuscript is now with our production department and you will be notified of the publication date in due course.

With kind regards,

Anita Estes

PLOS Genetics

On behalf of:
